# PKA drives an increase in AMPA receptor unitary conductance during LTP in the hippocampus

Pojeong Park[1,2,3,4], John Georgiou [3], Thomas M. Sanderson [1,2,3], Kwang-Hee Ko[2], Heather Kang[1,2,3,4], Ji-il Kim[2], Clarrisa A. Bradley [2,5], Zuner A. Bortolotto [1], Min Zhuo [2,4], Bong-Kiun Kaang [2] & Graham L. Collingridge [1,2,3,4,6✉]

Long-term potentiation (LTP) at hippocampal CA1 synapses can be expressed by an increase either in the number (N) of AMPA (α-amino-3-hydroxy-5-methyl-4-isoxazole propionic acid) receptors or in their single channel conductance (γ). Here, we have established how these distinct synaptic processes contribute to the expression of LTP in hippocampal slices obtained from young adult rodents. LTP induced by compressed theta burst stimulation (TBS), with a 10 s inter-episode interval, involves purely an increase in N (LTP$_N$). In contrast, either a spaced TBS, with a 10 min inter-episode interval, or a single TBS, delivered when PKA is activated, results in LTP that is associated with a transient increase in γ (LTP$_\gamma$), caused by the insertion of calcium-permeable (CP)-AMPA receptors. Activation of CaMKII is necessary and sufficient for LTP$_N$ whilst PKA is additionally required for LTP$_\gamma$. Thus, two mechanistically distinct forms of LTP co-exist at these synapses.

[1] Glutamate Receptor Group, School of Physiology, Pharmacology and Neuroscience, University of Bristol, Bristol, United Kingdom. [2] Department of Biological Sciences, College of Natural Sciences, Seoul National University, Seoul 08826, Korea. [3] Lunenfeld-Tanenbaum Research Institute, Mount Sinai Hospital, Toronto, ON M5G 1X5, Canada. [4] Department of Physiology, University of Toronto, Toronto, ON M5S 1A8, Canada. [5] Neurosciences and Mental Health, Peter Gilgan Centre for Research and Learning, The Hospital for Sick Children, Toronto, ON M5G 0A4, Canada. [6] TANZ Centre for Research in Neurodegenerative Diseases, University of Toronto, Toronto, ON M5S 1A8, Canada. ✉email: collingridge@lunenfeld.ca

Long-term potentiation (LTP) of synaptic function is considered the major process underlying learning and memory[1] where it is involved in synaptic engram formation[2,3], yet the underlying cellular mechanisms remain incompletely understood. The best-characterized form of LTP occurs at the Schaffer collateral-commissural pathway (SCCP) in the hippocampus, where it is triggered by synaptic activation of NMDA (N-methyl-D-aspartate) receptors[4] and is expressed as a persistent increase in AMPA (α-amino-3-hydroxy-5-methyl-4-isoxazole propionic acid) receptor-mediated synaptic transmission[5]. This modification is primarily due to a functional modulation of AMPA receptors (AMPARs), which may involve a change in the number of active channels (N) (termed $LTP_N$) and/or their single-channel conductance ($\gamma$) properties (termed $LTP_\gamma$) (e.g.[6–9]). Whilst there is considerable evidence that $LTP_N$ is triggered by activation of $Ca^{2+}$/calmodulin-dependent kinase II (CaMKII)[10,11] and involves exocytosis and lateral diffusion of AMPARs[12,13], the mechanisms underlying $LTP_\gamma$ are largely unknown. The two most likely molecular mechanisms involve (i) CaMKII-mediated phosphorylation of Ser831 of GluA1, which can result in an increase in the time AMPARs dwell in higher conductance states[14–16] or (ii) the insertion of calcium-permeable AMPA receptors (CP-AMPARs), which have a higher $\gamma$ than their calcium-impermeable (CI) counterparts[17,18].

In the present study, we have tested the hypothesis that $LTP_\gamma$ is due to the insertion of CP-AMPARs in young adult rodents using two theta burst stimulation (TBS) induction protocols that differed only in the timing between episodes, and applied peak-scaled non-stationary fluctuation analysis (NSFA)[19–21] to estimate $\gamma$ before and after the induction of LTP[6,15,22–25]. We found that the compressed TBS protocol (cTBS – inter-episode interval of 10 s) resulted exclusively in $LTP_N$, for which CaMKII was both necessary and sufficient. In contrast, a spaced TBS protocol (sTBS – inter-episode interval of 10 min) resulted in a transient increase in $\gamma$, lasting ~15 min, which was due to the insertion of CP-AMPARs and required both CaMKII and PKA. Insertion of CP-AMPARs mediates both the initial expression of $LTP_\gamma$, by enhancing the net synaptic unitary conductance, and helps trigger the processes that lead to a persistent increase in synaptic efficacy that outlasts the increase in $\gamma$. Since the PKA-dependent form of LTP also requires de novo protein synthesis and has stimulation features similar to spaced behavioural learning, $LTP_\gamma$ is likely to underlie the formation of synaptic engrams and long-term memory.

## Results

**An increase in $\gamma$ is specifically triggered by a sTBS protocol.** Simultaneous field excitatory postsynaptic potential (fEPSP) recordings from stratum radiatum and somatic whole-cell recordings were obtained in response to baseline stimulation of two independent SCCP inputs (Fig. 1a). TBS was delivered to one input (test), while the second input served as a control for stability and heterosynaptic effects (Fig. 1c, d). Synaptic potentiation was quantified and $\gamma$ was estimated using NSFA (Fig. 1e, f), as described previously[6]. To optimize the estimates of $\gamma$ we used minimal stimulation and restricted our measurements to the first 20–30 min following TBS, since $\gamma$ estimates are extremely sensitive to small fluctuations in series resistance[20]. Thus, our study focused on the induction and initial expression mechanisms of LTP.

In the first series of experiments we delivered three episodes of TBS, with each episode comprising 5 shocks at 100 Hz delivered 5 times at 5 Hz (i.e. 75 stimuli in total; see Fig. 1b schematic); in interleaved experiments we either delivered these three episodes as a cTBS (10 s inter-episode interval) or as a sTBS (10 min inter-

episode interval). We referred to the resultant potentiation as cLTP (Fig. 2a–i) and sLTP (Fig. 2j–r), respectively. In response to cTBS there was a substantial cLTP (Fig. 2a), with EPSC amplitudes increasing to 212 ± 11% of baseline, averaged over the first 10 min after induction (Fig. 2b). For 22 neurons from 15 rats ($n = 22/15$), we obtained $\gamma$ estimates in 10 min epochs and found it to be unaltered throughout (Fig. 2c–g). The $\gamma$ values were 5.1 ± 0.3 pS, (baseline), 5.3 ± 0.4 pS (first 10 min epoch post cTBS; $LTP_{10}$; $t_{21} = 1.23$, $p = 0.2327$, vs baseline, paired Student's $t$ test) and 5.2 ± 0.4 pS (second 10 min epoch post cTBS; $LTP_{20}$; $t_{21} = 0.33$, $p = 0.7452$; Fig. 2d). The control input was also stable throughout (4.9 ± 0.4 pS, 4.5 ± 0.3 pS and 4.8 ± 0.3 pS at the corresponding time-points; Fig. 2d). The lack of change in $\gamma$ was also clearly evident in the plots from individual experiments for control (Fig. 2e) and test inputs (Fig. 2f) and in the cumulative distribution plots (Fig. 2g). The lack of change in $\gamma$ was observed over a wide range of cLTP magnitudes (Fig. 2h).

In response to sTBS the results were strikingly different. For this set of experiments, whole-cell recordings were obtained shortly after delivery of the second TBS episode and the effects of the third TBS were evaluated (Fig. 2j). This method was necessary because of the rapid wash-out of LTP with low access whole-cell recordings. In response to the third TBS there was a substantial additional LTP, with EPSC amplitudes increasing to 177 ± 9% of baseline, averaged over the first 10 min after induction (Fig. 2k). The estimate of $\gamma$ upon break in was significantly higher (6.9 ± 0.4 pS) compared to the control input (4.9 ± 0.4 pS; Fig. 2n–o; $t_{22} = 3.22$, $p = 0.0039$, paired Student's $t$ test) and this was further increased in response to the third episode of TBS to 8.4 ± 0.4 pS ($LTP_{10}$; $t_{22} = 3.75$, $p = 0.0011$, Fig. 2l, m, o, p; $n = 23/17$). However, when we quantified $\gamma$ at 10–20 min after the last TBS, the value (5.5 ± 0.3 pS) was no longer significantly different from the control input ($LTP_{20}$; $t_{22} = 2.01$, $p = 0.0570$, paired Student's $t$ test; Fig. 2m). In contrast to the test input, sTBS did not result in a significant $\gamma$ change in the control input (4.9 ± 0.4 pS, 5.4 ± 0.4 pS and 4.6 ± 0.3 pS at the corresponding time points; Fig. 2m, n). Thus, the increase in $\gamma$ is specifically related to sLTP. Furthermore, this increase in $\gamma$ positively correlated with the magnitude of sLTP (Fig. 2q).

Since sLTP, but not cLTP, is associated with the insertion of CP-AMPARs[26,27] these results suggest that CP-AMPARs may account for the increase in $\gamma$. CP-AMPARs have slightly faster decay kinetics ($\tau_{decay}$) than CI-AMPARs[25,28], which can be detected using single exponential fits to EPSC decays. We found that cLTP was not associated with an alteration in $\tau_{decay}$ (Fig. 2i, Supplementary Table 1; $t_{21} = 0.66$, $p = 0.5146$, paired Student's $t$ test), whereas sLTP was associated with a highly significant decrease in $\tau_{decay}$ (Supplementary Table 1; $p = 0.0051$, $t_{22} = 3.11$, paired Student's $t$ test). A regression analysis showed a trend for the $\tau_{decay}$ to be inversely related with the increase in $\gamma$ (Fig. 2r; $p = 0.0712$, $F_{(1,21)} = 3.61$). Therefore, the kinetic analysis provides additional support for the notion that insertion of CP-AMPARs occurs during the induction of LTP in response to a sTBS.

**The role of PKA in $LTP_\gamma$.** It is established that elevating cAMP by, for example, use of the phosphodiesterase 4 inhibitor rolipram, enables a weak stimulus to generate an enhanced PKA-dependent form of LTP[29]. Previously, we found that in the presence of rolipram a weak TBS, comprising one episode of TBS, generated an LTP that is largely dependent on the insertion of CP-AMPARs[26]. Here we used this same method as an independent means to investigate whether insertion of CP-AMPARs are responsible for the increase in $\gamma$. Since only one TBS is required to induce the PKA-dependent form of LTP in the presence of rolipram we could make $\gamma$ measurements before and after the full

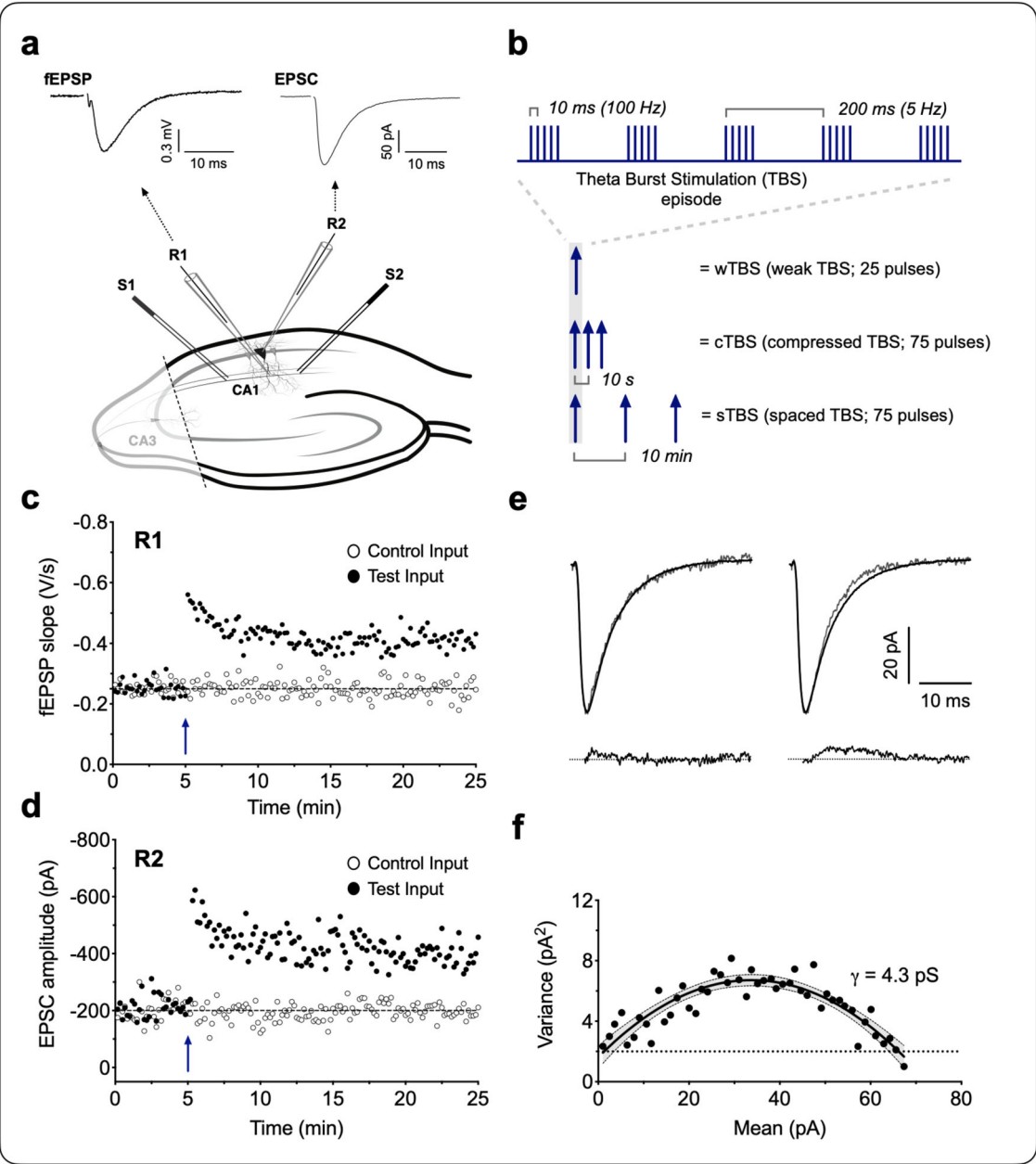

**Fig. 1 LTP and non-stationary fluctuation analysis (NSFA) methodology. a** Schematic of a hippocampal brain slice for LTP experiments, and the positioning of recording (R1, R2) and stimulating (S1, S2) electrodes. The CA3 region was cut (dashed line) to reduce neuronal excitability. Representative field and whole cell responses (fEPSP and EPSC), simultaneously obtained from CA1 neurons. Five consecutive responses were averaged and the stimulus artifacts were blanked for clarity. **b** Induction protocols for weak, compressed and spaced TBS (wTBS, cTBS and sTBS) are graphically summarized. **c** Representative fEPSP recordings for LTP evoked by a single episode of TBS (weak TBS, blue arrow). **d** Simultaneously obtained EPSC recordings. **e** Upper traces are two sets of representative waveforms for individual sweeps (thin lines), superimposed with the scaled mean of 57 EPSCs (thick lines). Lower traces are the subtraction of the scaled mean from the representative individual EPSCs. **f** Corresponding current–variance relationship to estimate the unitary conductance ($\gamma$). Fluctuation of the individual decays was plotted against the mean EPSC. Solid line is a parabolic fit with 95% confidence intervals (shaded). Dotted line, the background average variance.

induction of LTP. As illustrated in Fig. 3a, b, a single episode of TBS (wTBS; comprising 25 stimuli), when delivered in the presence of rolipram (1 µM), generated a robust LTP (234 ± 14% of baseline for test vs. 121 ± 6% for control input). We found that this LTP was also associated with a transient increase in $\gamma$ (baseline = 4.9 ± 0.4 pS, $LTP_{10'}$ = 8.0 ± 0.6 pS; $t_{20}$ = 5.90, $p <$ 0.0001) that returned to baseline by the second 10 min epoch ($LTP_{20'}$ = 5.4 ± 0.3 pS; $t_{20}$ = 1.39, $p$ = 0.1810, paired Student's $t$ test) following the wTBS ($n$ = 21/15; Fig. 3c, d, f, g). This potentiation required the wTBS since the control input was

largely unaffected (5.1 ± 0.3 pS, 5.4 ± 0.5 pS and 4.8 ± 0.3 pS at the corresponding time points; Fig. 3d, e) and since the baseline $\gamma$ values in the presence of rolipram were not significantly different to the baseline $\gamma$ values in its absence (Fig. 3d–f; Supplementary Table 1). As was the case with the sLTP, the size of the change in $\gamma$ correlated with the magnitude of LTP ($p$ = 0.0024, $F_{(1,19)}$ = 12.27; Fig. 3h). Additionally, there was an associated reduction in $\tau_{decay}$ ($p$ = 0.0007, $t_{20}$ = 3.99, paired Student's $t$ test; Supplementary Table 1) that also negatively correlated with the increased $\gamma$ ($p$ = 0.0199, $F_{(1,19)}$ = 6.46; Fig. 3i). These results

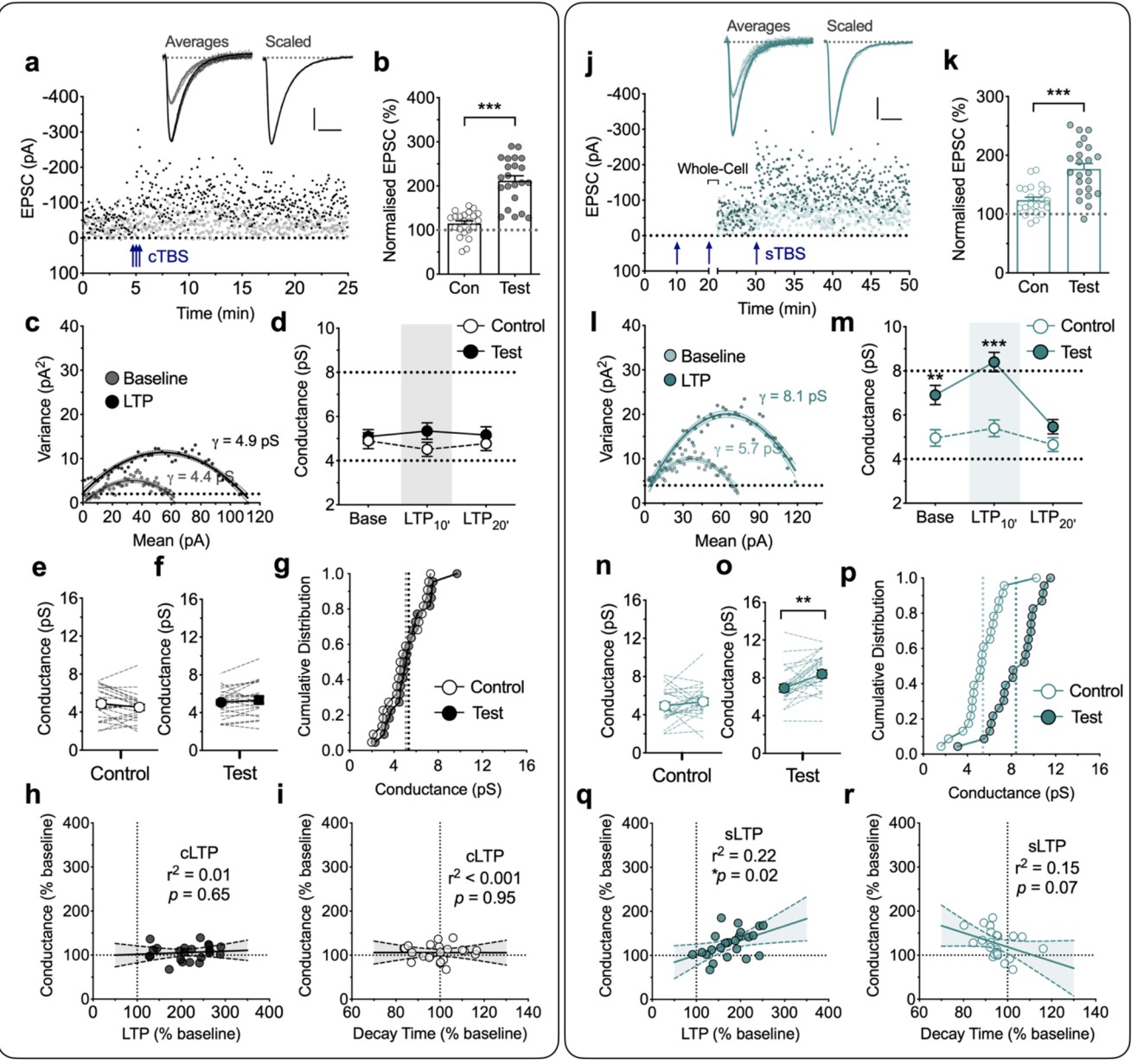

**Fig. 2 Increased AMPA receptor unitary conductance (γ) during sLTP, but not cLTP. a** A representative LTP experiment with sample traces for baseline and post TBS – the mean of selected records for analysis, superimposed with peak-scaled individual traces (10 successive sweeps, thin lines; baseline = grey, LTP = black). Scaled trace is from the baseline normalized to the LTP. Scale bars: 20 pA and 10 ms. Two inputs were stimulated alternately and cTBS (3 x TBS with an inter-episode interval of 10 s; blue arrows) delivered to one input (filled symbols) with the second input (open symbols) serving as a control (Con). **b** Levels of cTBS-induced LTP (cLTP) for control and test inputs, quantified during the 10 min epoch after the induction (mean ± SEM, $n = 22$ neurons from 15 animals; $t_{21} = 8.545$, $p < 0.0001$, two-sided paired Student's $t$ test). **c** Corresponding current–variance relationship of the EPSCs for the test input. The unitary channel conductance (γ) of AMPA receptors was estimated during baseline (grey) and after the induction of LTP (LTP$_{10'}$; black). **d** Grouped comparison of control and test input γ estimates for baseline and the initial 10 min epoch (LTP$_{10'}$) and the subsequent 10 min epoch (LTP$_{20'}$). $n = 22$ neurons from 15 animals. **e, f** Summary plot for the γ at baseline (left) and LTP$_{10'}$ (right) for control (**e**) and test (**f**) inputs. Individual values from each neuron are connected by lines. Circles indicate mean values. **g** Cumulative distribution of the same data set for LTP$_{10'}$. Dotted lines indicate the mean values for each input. **h, i** Analysis of the relationships of γ with LTP ($p = 0.6517$, $F_{(1,20)} = 0.2101$, $F$-test) (**h**) and EPSC decay time ($p = 0.9521$, $F_{(1,20)} = 0.0037$, $F$-test) (**i**). Linear regression with 95% confidence intervals (shaded) for the amount of cLTP and the corresponding level of γ. **j–r** Equivalent analysis for the LTP induced by sTBS (3 x TBS at inter-episode interval of 10 min; see arrows). The whole-cell recordings were obtained after the second TBS. This was necessary due to the lability of LTP washout. **k** Levels of sTBS-induced LTP (sLTP) for control and test inputs, quantified during the 10 min epoch after the induction ($n = 23$ neurons from 17 animals; $t_{22} = 5.238$, $p < 0.0001$, two-sided paired Student's $t$ test). **m–o** Statistical analysis between control and test pathways ($t_{22} = 3.220$, $p = 0.0039$ for baseline and $t_{22} = 6.123$, $p < 0.0001$ for LTP$_{10'}$, two-sided paired Student's $t$ test) (**m**) and within pathway analysis for control ($t_{22} = 1.065$, $p = 0.2986$, two-sided paired Student's $t$ test) (**n**) and test ($t_{22} = 3.753$, $p = 0.0011$, two-sided paired Student's $t$ test) (**o**) pathway reveals a time- and pathway-dependent increase in γ. Note that higher conductance was observed in the test input (**o**) compared to the control (**n**) under the "baseline" state, suggesting that the first + second TBS were sufficient to increase γ. The third TBS triggered a small but discernible further increase in γ. **q, r** Analysis of the relationships of γ with LTP ($p = 0.0225$, $F_{(1, 21)} = 6.066$, $F$-test) (**q**) and decay time of EPSCs ($p = 0.0712$, $F_{(1,21)} = 3.612$, $F$-test) (**r**). Data are presented as mean ± SEM. Source data are provided as a Source Data file.

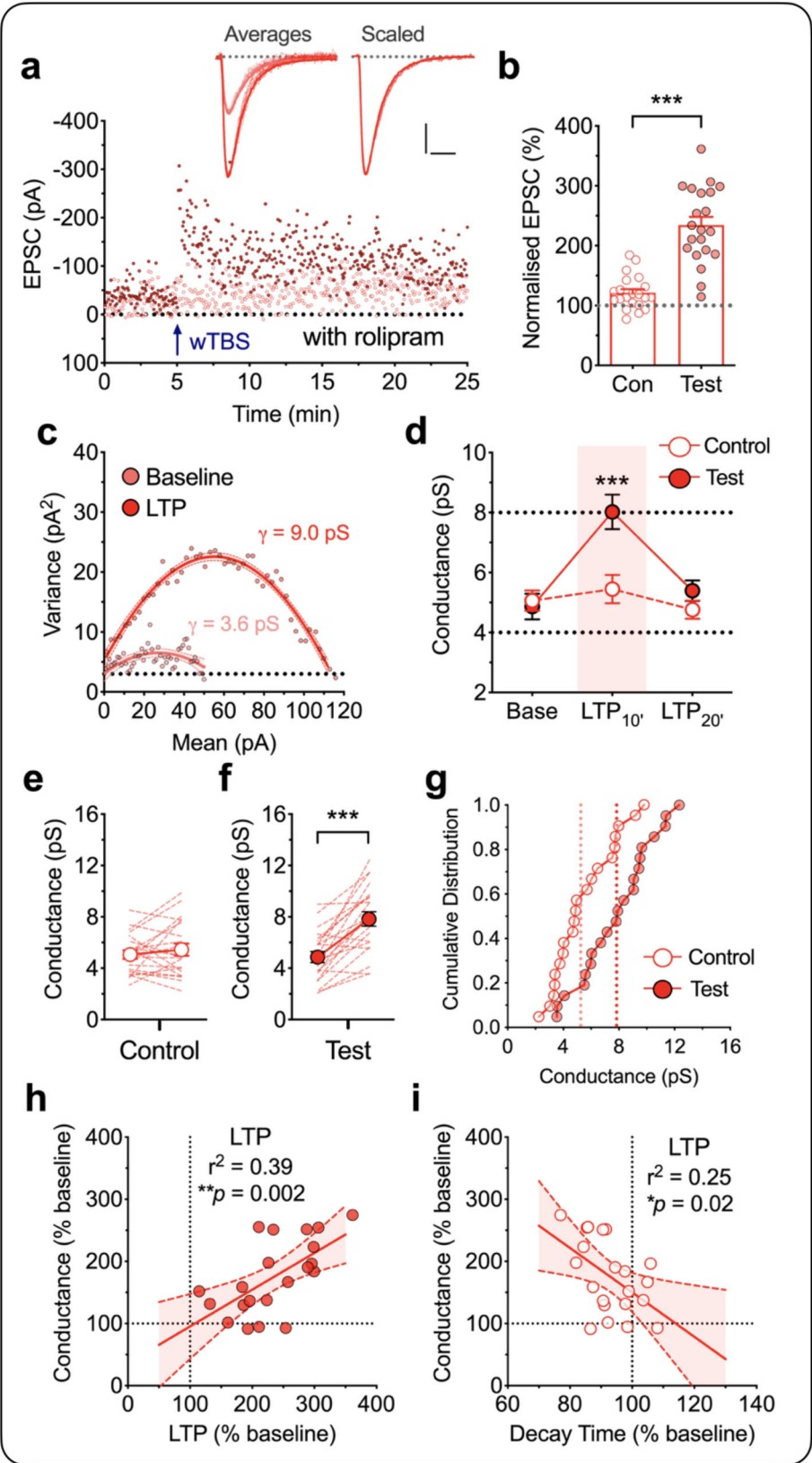

further support the idea that insertion of CP-AMPARs mediates LTP$_\gamma$.

To more specifically test the requirement of PKA for driving alterations in $\gamma$, we included the catalytic subunit of PKA (PKA C$\alpha$; 300 U/mL) in the patch solution (Fig. 4). This treatment had little effect on the control input that did not receive any wTBS

(Fig. 4a), suggesting that PKA alone has minimal effect on synaptic transmission. However, as was the case with rolipram, the wTBS in the presence of PKA C$\alpha$ generated a robust potentiation (Fig. 4a) that was associated with an increase in $\gamma$ (Fig. 4b, c). The levels quantified during baseline and 10 min post TBS (LTP$_{10'}$) were $5.2 \pm 0.5$ pS and $7.8 \pm 0.8$ pS ($t_{16} = 5.80$, $p <$

**Fig. 3 Increased AMPA receptor unitary conductance ($\gamma$) during LTP in the presence of rolipram. a–g** Equivalent experiments to those illustrated in Fig. 2 for the LTP induced by a wTBS (a single episode of TBS) in the presence of rolipram (1 μM; $n = 21$ neurons from 15 animals). Scale bars: 10 pA and 10 ms. **b** Quantification for the levels of LTP for control and test inputs during the 10 min epoch after the induction (mean ± SEM, $t_{20} = 7.860$, $p < 0.0001$, two-sided paired Student's $t$ test). **d–f** Statistical analysis between control and test pathways for LTP$_{10'}$ ($t_{20} = 5.901$, $p < 0.0001$, two-sided paired Student's $t$ test) (**d**) and within pathway analysis for control ($t_{20} = 0.4416$, $p = 0.6635$, two-sided Student's $t$ test) (**e**) and test ($t_{20} = 6.059$, $p < 0.0001$, two-sided paired Student's $t$ test) (**f**) inputs. **h**, **i** Analysis of the relationships of $\gamma$ with LTP ($p = 0.0024$, $F_{(1,19)} = 12.27$, $F$-test) (**h**) and decay time of EPSCs ($p = 0.0199$, $F_{(1,19)} = 6.462$, $F$-test) (**i**). Data are presented as mean ± SEM. Source data are provided as a Source Data file.

0.0001, paired Student's $t$ test; $n = 17/13$; Fig. 4b). Once again, the increase in $\gamma$ was only transient, since estimates of $\gamma$ made between 10 and 20 min following the wTBS (i.e. LTP$_{20'}$) were not significantly different from baseline ($5.3 \pm 0.5$ pS; $t_{16} = 0.37$, $p = 0.7163$, paired Student's $t$ test; Fig. 4b).

To establish whether the increase in $\gamma$ was indeed due to the insertion of CP-AMPARs we used IEM-1460 (IEM, 30 μM). Previously, we showed that IEM inhibited LTP triggered by a sTBS without affecting LTP triggered by a cTBS[26,27]. Since these two induction protocols activate NMDARs to a similar extent, the effects of IEM is unlikely to be due to a direct action on NMDARs. To establish whether this is indeed the case, we examined the effects of IEM on NMDAR-mediated EPSCs evoked by single pulses and during TBS. IEM had no effect whatsoever on NMDAR-mediated synaptic transmission (Supplementary Fig. 1).

In the presence of bath applied IEM and PKA Cα in the patch pipette, the level of LTP triggered by the wTBS was significantly less than in its absence ($202 \pm 16\%$ vs. $276 \pm 19\%$ of baseline, 10 min after wTBS; $t_{31} = 3.01$, $p = 0.0052$, unpaired Student's $t$ test; Fig. 4d, g), consistent with a component of LTP being generated by the insertion of CP-AMPARs when PKA is activated[26,30–32]. IEM completely prevented the transient increase in $\gamma$ (baseline vs. LTP$_{10'}$; $4.3 \pm 0.5$ pS vs. $4.3 \pm 0.5$ pS; $t_{15} = 0.08$, $p = 0.9338$, paired Student's $t$ test; Fig. 4e, h; $n = 16/13$; also see Supplementary Table 1). There was a strong correlation between the increase in $\gamma$ with both the magnitude of LTP (Fig. 4i; $p = 0.0021$, $F_{(1,15)} = 13.72$) and the decrease in $\tau_{decay}$ (Fig. 4k; $p = 0.0117$, $F_{(1,15)} = 8.24$) when wTBS was delivered in the presence of PKA Cα, but there were no such correlations when IEM was also present (Fig. 4j, l).

In conclusion, we find that activation of PKA, that occurs during (i) a sTBS, (ii) a wTBS in the presence of rolipram or (iii) a wTBS in the presence of the catalytic subunit of PKA, results in the transient insertion of CP-AMPARs and that these receptors are responsible for the increase in $\gamma$ during the initial expression phase of LTP.

**The role of CaMKII in LTP$_\gamma$.** CaMKII has been demonstrated to be both necessary and sufficient for the induction of LTP[10,11]. Consistent with this notion, when tested using a CaMKII selective antagonist, KN-62 (10 μM), we found that both cLTP and sLTP were substantially reduced (Fig. 5a, b). The levels of potentiation of $108 \pm 8\%$ (after 90 min of cTBS, $n = 4$ slices from individual animals; Fig. 5a) and $105 \pm 7\%$ (after 120 min of sTBS, $n = 5$ slices; Fig. 5b), respectively, were significantly less than that in the corresponding interleaved untreated control groups that potentiated $155 \pm 5\%$ ($p = 0.0010$, $t_9 = 4.79$; unpaired Student's $t$ test; $n = 7$ slices) and $159 \pm 3\%$ ($p = 0.0001$, $t_{13} = 6.87$; unpaired Student's $t$ test; $n = 10$ slices), but were not significantly different from their respective control inputs ($t_3 = 1.50$, $p = 0.2315$ and $t_4 = 0.66$; $p = 0.5426$; paired Student's $t$ test).

It has been suggested that the role of CaMKII in LTP involves an increase in $\gamma$[14,15]. To further examine the role of CaMKII in LTP we interleaved experiments where we applied either active or inactive (heat inactivated) CaMKII (250 U/mL) via the patch pipette and delivered baseline (low frequency) stimulation to

monitor basal synaptic transmission. Consistent with previous reports[33,34], activated CaMKII, but not inactive CaMKII, was sufficient to potentiate synaptic transmission (Fig. 5c, d, h). However, this potentiation was not associated with an increase in $\gamma$ (Fig. 5f, g, i) or a change in rise and decay kinetics (Fig. 5e; see also Supplementary Table 1). The respective $\gamma$ values for baseline (i.e. first 5 min of recording) and 10–15 min of whole-cell recording were $4.7 \pm 0.6$ pS and $4.3 \pm 0.5$ pS ($t_{14} = 0.74$, $p = 0.4740$, paired Student's $t$ test; $n = 15/12$; Fig. 5g). There was no correlation between $\gamma$ change and either the magnitude of LTP (Fig. 5j; $p = 0.2265$, $F_{(1,13)} = 1.61$) or $\tau_{decay}$ (Fig. 5k; $p = 0.2813$, $F_{(1,13)} = 1.26$). We can conclude, therefore, that CaMKII alone can generate substantial potentiation that does not involve any alteration in $\gamma$.

**Activation of CaMKII and PKA are both necessary and sufficient for LTP$_\gamma$.** Since neither PKA alone nor CaMKII alone affected $\gamma$, we explored whether the combination of the two kinases may be sufficient for the effect. We, therefore, patch loaded PKA Cα (300 U/mL) with either the active or inactive forms of CaMKII (250 U/mL). In interleaved experiments, we found that PKA Cα + active CaMKII produced a robust potentiation of synaptic responses, specifically $178 \pm 10\%$ of baseline when quantified 15 min after whole-cell (Fig. 6a, b, f). In this case, the effect was also associated with an increase in $\gamma$ (Fig. 6d, e, g). The levels of conductance for the baseline and potentiation (calculated between 10 to 15 min of recording) were $4.6 \pm 0.4$ pS and $6.5 \pm 0.4$ pS, respectively ($t_{17} = 5.38$, $p = 0.0002$, paired Student's $t$ test; $n = 18/15$). Again, this effect was only transient, as the $\gamma$ returned to baseline levels within 20–30 min of whole-cell recording (Fig. 6e). In contrast, inactive CaMKII plus PKA Cα, had no significant effect on synaptic transmission ($112 \pm 9\%$; Fig. 6b, f) or on $\gamma$ ($4.2 \pm 0.4$ pS vs. $4.2 \pm 0.4$ pS, $n = 16/14$; Fig. 6e, g). These results suggest that (i) both CaMKII and PKA are required for and (ii) their combined activity is sufficient for LTP$_\gamma$ at these synapses.

In additional interleaved experiments, the sensitivity to IEM was tested on the potentiation produced by CaMKII plus PKA Cα. Consistent with the involvement of CP-AMPARs, there was a reduced level of potentiation (Fig. 6b, f) and no change in $\gamma$ in the presence of IEM (Fig. 6e, g). The respective amounts, quantified after 10–15 min of whole-cell recording, were $128 \pm 9\%$ of baseline ($p = 0.0003$ vs. CaMKII + PKA Cα, one-way ANOVA with Bonferroni's correction) and $4.1 \pm 0.4$ pS ($p = 0.0009$ vs. CaMKII + PKA Cα, one-way ANOVA with Bonferroni's correction; $n = 20/15$).

The correlations between the change in $\gamma$ with the level of LTP and decay times for these experiments are summarized in Fig. 6h–k. There was a substantial correlation between the increase in $\gamma$ and the level of potentiation (Fig. 6h) and the decrease in $\tau_{decay}$ (Fig. 6i) with CaMKII plus PKA Cα but no such correlation was found in the presence of IEM (Fig. 6j, k).

**The proportion of synaptically incorporated CP-AMPARs during LTP$_\gamma$.** Together, the previous experiments provide multiple lines of evidence that LTP$_\gamma$ is due to the insertion of CP-AMPARs

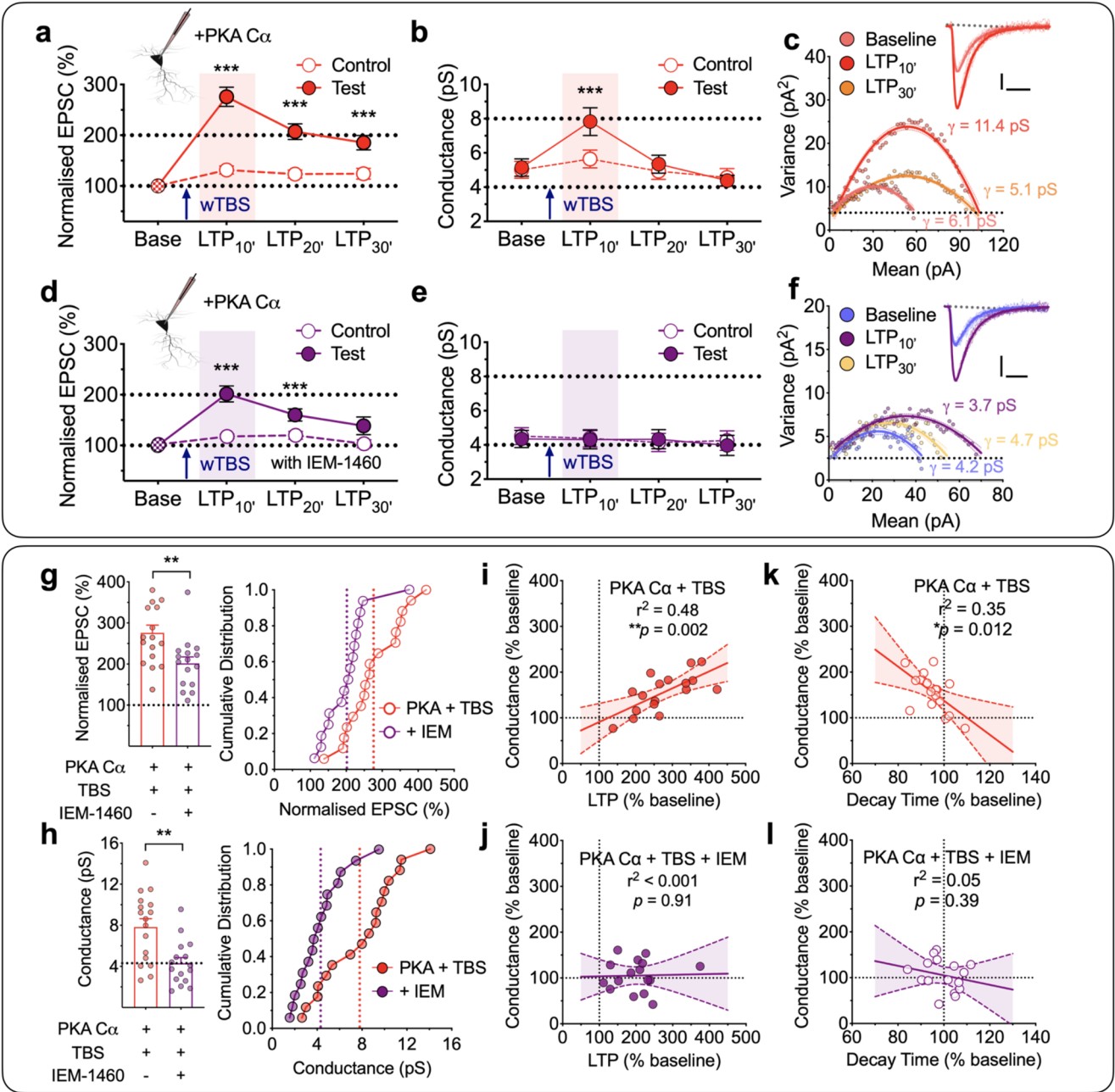

**Fig. 4 wTBS with PKA Cα transiently increases γ via CP-AMPAR insertion. a–b** A wTBS in the presence of intracellular PKA Cα (300 U/mL) transiently increased γ ($n = 17$ neurons from 13 animals, mean ± SEM, $F_{(1.931, 27.04)} = 52.89$ for EPSCs and $F_{(1.863, 26.08)} = 12.59$ for γ, one-way repeated measures ANOVA followed by Bonferroni's multiple comparisons test; $*p < 0.05$, $**p < 0.01$ and $***p < 0.001$ vs. baseline). EPSCs (**a**) and γ (**b**) were analyzed in 10-min bins. A single episode of TBS (at time marked by an arrow) was delivered to one input (filled symbols) with the second input (open symbols) serving as a control; base = baseline. **c** A representative current–variance plot for PKA Cα plus wTBS for baseline, the first 10 min (LTP$_{10'}$) and the last 10 min of LTP (LTP$_{30'}$). Sample traces were obtained from baseline and LTP$_{10'}$. Scale bars: 10 pA and 10 ms. **d–f** Equivalent experiments in the presence of IEM-1460 (IEM, 30 μM; $n = 16$ neurons from 13 animals, $F_{(1.095, 13.14)} = 25.66$ for EPSCs and $F_{(2.184, 26.21)} = 0.2547$ for γ, one-way repeated measures ANOVA followed by Bonferroni's multiple comparisons test; $*p < 0.05$, $**p < 0.01$ and $***p < 0.001$ vs. baseline). **g, h** Quantification of the levels of LTP ($t_{31} = 3.006$, $p = 0.0052$, two-sided unpaired Student's t test) (**g**) and ($t_{31} = 3.544$, $p = 0.0013$, two-sided unpaired Student's t test) γ (**h**) measured during the 10 min after wTBS with cumulative distributions (right). $n = 17$ neurons from 13 animals (PKA Cα + wTBS) and 16 neurons from 13 animals (PKA Cα + wTBS + IEM). **i, j** Analysis of the relationships between γ and LTP for PKA Cα + wTBS ($p = 0.0021$, $F_{(1,15)} = 13.72$, F-test) (**i**) and PKA Cα + wTBS + IEM ($p = 0.9090$, $F_{(1,14)} = 0.0136$, F-test) (**j**). **k, l** Analysis of the relationships between γ and EPSC decay time for PKA Cα + wTBS ($p = 0.0117$, $F_{(1,15)} = 8.243$, F-test) (**k**) and PKA Cα + wTBS + IEM ($p = 0.3931$, $F_{(1,14)} = 0.7764$, F-test) (**l**). Data are presented as mean ± SEM. Source data are provided as a Source Data file.

into synapses that contain CI-AMPARs. In order to determine the relative proportions of each it was necessary to measure γ for synapses containing either just CI-AMPARs or just CP-AMPARs, under our recording conditions. To achieve this, we used lentivirus-driven CRISPR/Cas9 expression to delete GluA2 in a fraction of

neurons in vivo, allowing a direct comparison between a knock-out (KO) and a wild-type (WT) neuron within each adult brain slice (Fig. 7a). When compared with uninfected neighbouring neurons, the KO cells showed a reduced AMPAR synaptic transmission (Fig. 7b) and an inwardly rectifying current–voltage relationship

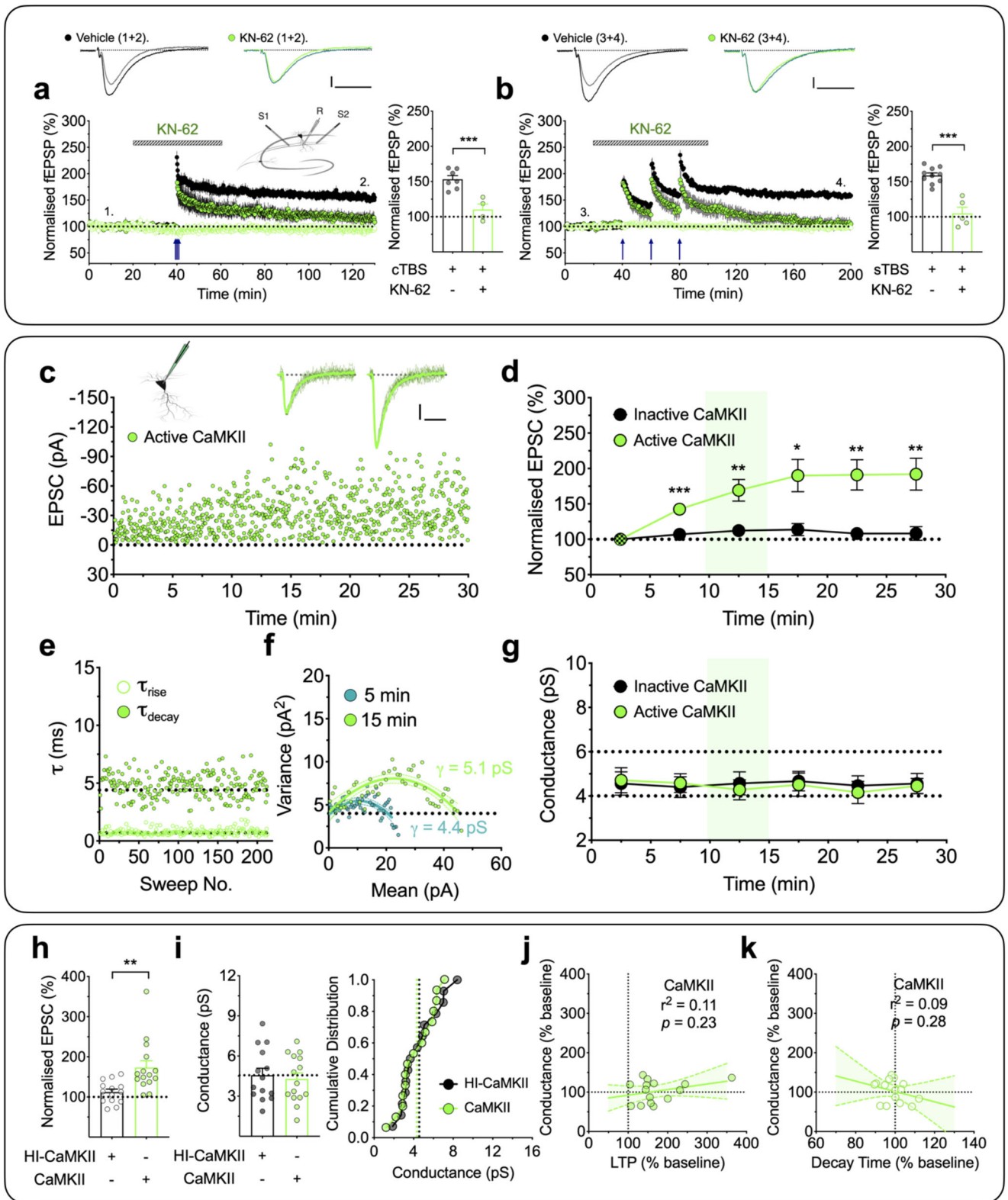

(Fig. 7c, d). The level of $\gamma$ in KO neurons was significantly higher at $17.3 \pm 1.2$ pS ($n = 16$) compared to $4.6 \pm 0.4$ pS for WT neurons ($n = 17$; $t_{31} = 10.09$, $p < 0.0001$, unpaired Student's $t$ test, data from 12 animals; Fig. 7e–h). This increase in $\gamma$ in these neurons was associated with a decreased $\tau_{decay}$ from $6.8 \pm 0.4$ ms in WT to $5.3 \pm 0.4$ ms in KO neurons ($t_{31} = 2.85$, $p = 0.0026$, unpaired Student's $t$ test, Fig. 7i). Assuming that these EPSCs were comprised of 100 and 0 % CP-AMPARs, respectively, then the increase in $\gamma$ that we observed during

sLTP can be explained by CP-AMPARs comprising ~30% of the synaptic current during the first 10 min following LTP induction.

## Discussion

NMDA receptor-dependent LTP has been extensively studied as the primary mechanisms utilized are crucial for the formation of long-term memories. Despite many molecules being discovered

**Fig. 5 CaMKII does not affect $\gamma$. a, b** CaMKII dependence of both forms of LTP (mean ± SEM). **a** cLTP, measured using fEPSP recordings, was inhibited by the CaMKII inhibitor, KN-62 (10 µM; $n = 4$ animals; green). **b** sLTP showed a similar sensitivity to KN-62 ($n = 5$ animals). Interleaved control experiments ($n = 7$ and 10 animals; black) are superimposed. $t_9 = 4.786$ ($p = 0.0010$; cTBS) and $t_{13} = 6.865$, ($p < 0.0001$; sTBS) by two-sided unpaired Student's $t$ test. The sample traces were obtained at the time indicated by the numbers. Scale bars: 0.2 mV and 10 ms. **c** A representative whole-cell recording with the inclusion of activated CaMKII (250 U/mL) in the internal solution. The sample traces are averages of selected records for analysis, superimposed with individual scaled traces (10 successive sweeps, thin lines) after 5 and 15 min of whole-cell recording. Scale bars: 10 pA and 10 ms. **d** Pooled results (mean ± SEM, 5-min bins) for the effects on EPSC (%) by activated CaMKII ($n = 15$ neurons from 12 animals, $F_{(1.378, 19.30)} = 10.52$, $p = 0.0021$, one-way repeated measures ANOVA) and interleaved control, heat-inactivated, CaMKII ($n = 14$ neurons from 11 animals, $F_{(2.585, 33.61)} = 1.096$, one-way repeated measures ANOVA). Bonferroni's post hoc multiple comparisons test; *$p < 0.05$, **$p < 0.01$ and ***$p < 0.001$ vs. the initial 5 min of whole-cell recording. **e** Rise times (20–80%, $\tau_{rise}$) and decay time constants ($\tau_{decay}$) were plotted for the EPSCs used in the NSFA analysis for the neuron illustrated in **c**. **f** Current–variance relationships for this neuron used to estimate $\gamma$ over 5 min epochs starting at 0 and 10 min after commencing whole-cell recording. **g** Time course for the estimates of $\gamma$ for active vs. inactive CaMKII. One-way repeated measures ANOVA; $F_{(3.837, 53.72)} = 0.2498$ ($n = 15$ neurons from 12 animals, active CaMKII), $F_{(3.706, 48.18)} = 0.0795$ ($n = 14$ neurons from 11 animals, inactive CaMKII). Bonferroni's post hoc multiple comparisons test; *$p < 0.05$, **$p < 0.01$ and ***$p < 0.001$ vs. the initial 5 min of whole-cell recording. **h, i** Quantification for the levels of LTP ($t_{27} = 3.125$, $p = 0.0040$, two-sided unpaired Student's $t$ test) (**h**) and $\gamma$ ($t_{27} = 0.3539$, $p = 0.7262$, two-sided unpaired Student's $t$ test) (**i**) measured over a 5 min epoch, commencing 10 min after starting whole-cell recording. **j, k** Analysis of the relationships between $\gamma$ and LTP ($p = 0.2265$, $F_{(1,13)} = 1.611$, F-test) (**j**) and EPSC decay time ($p = 0.2813$, $F_{(1,13)} = 1.264$, F-test) (**k**) for the active CaMKII experiments. Data are presented as mean ± SEM. Source data are provided as a Source Data file.

and different aspects of their regulation being uncovered, there are crucial gaps in our knowledge. One relates to the fact that long-term memory requires de novo protein synthesis yet most of our mechanistic understanding of LTP has been obtained from the study of a protein synthesis-independent form of LTP. A second pertains to the fact that much of this understanding has been derived from the study of juvenile animals, where technical issues have permitted more in-depth analysis, whereas most studies of learning and memory are conducted in adult animals. In the present study, we have addressed these issues by studying LTP at CA1 synapses in young adult rodents and have compared induction protocols that are known to activate the protein synthesis-independent (cTBS) and protein synthesis-dependent (sTBS, rolipram + wTBS) forms[35,36]. Using a cTBS protocol, LTP involved the insertion of additional CI-AMPARs, for which activation of CaMKII is both necessary and sufficient. Using a sTBS there was an additional LTP component that involved the transient insertion of CP-AMPARs, for which activation of CaMKII and PKA are both necessary and, in combination, sufficient. The insertion of CP-AMPARs increases AMPA receptor $\gamma$ and this underlies the initial expression of this form of LTP, which we have termed LTP$_\gamma$. The insertion of CP-AMPARs is transient and is replaced by a persistent increase in the number of CI-AMPARs.

**Two distinct postsynaptic forms of LTP at CA1 synapses.** The division of NMDA receptor-dependent LTP into multiple components was made on the basis of sensitivity to various pharmacological agents and substantiated by genetic studies[36]. In particular, when a single train (tetanus or TBS) is employed, the resultant LTP may be independent of both PKA activation and de novo protein synthesis; this is commonly referred to as LTP1 or E-LTP[35,37]. In contrast, when multiple trains are delivered, with an interval in the order of minutes, then there is often the generation of an additional PKA and de novo protein synthesis-dependent component of LTP, which is commonly referred to as LTP2 or L-LTP[27,36,38,39]. LTP2 is generally assumed to underlie long-term memory formation, that also requires de novo protein synthesis.

NMDA receptor-dependent LTP has also been divided into two distinct postsynaptic mechanisms of expression, one involving an increase in the number of AMPARs without a change in $\gamma$ (LTP$_N$) and the other involving an increase in $\gamma$ (LTP$_\gamma$)[6]. Here, one of our goals was to determine whether these separate expression mechanisms specifically relate to LTP1 and LTP2. We found that LTP1 never involved an alteration in $\gamma$ whereas LTP2

invariably did. The increase in $\gamma$ was transient, lasting between 10 and 20 min and could be fully explained by the insertion of CP-AMPARs. In terms of signalling cascades, we found that activation of CaMKII was both necessary and sufficient for LTP1 whereas both CaMKII and PKA were required, and in combination were sufficient, for LTP2 (see model in Fig. 8). Our findings do not conflict with a large body of literature regarding alterations in AMPARs underlying LTP at these synapses and the roles of both CaMKII and PKA (e.g.[26,30,40–42]).

The roles of CP-AMPARs and alterations in $\gamma$ in LTP have been controversial[6,24,43–45]. However, these controversies can now be reconciled on the basis of the type of LTP under investigation. Under the conditions of our study, we could readily switch between forms of LTP that do not (LTP1) or do (LTP2) involve a CP-AMPAR component by simply altering the timing between TBS episodes. We saw similar effects when we compared whole-cell recordings, using minimal stimulation, with field potential recordings, which provide an average measure of synaptic transmission across a wide range of release probabilities $P(r)$. Therefore, we do not expect that our observations are dependent on the $P(r)$ of the synapse under investigation. However, the extent to which CP-AMPARs are involved in synaptic plasticity is likely to involve additional factors, such as the developmental stage of the animal, the level of stress experienced prior to euthanasia and the precise experimental conditions used, including the stimulus parameters employed[31,32,43,44,46,47].

We can conclude that LTP1 equates to LTP$_N$ and LTP2 with LTP$_\gamma$. It is important to note, however, that although a compressed induction protocol (cTBS) will ordinarily result in just LTP1/LTP$_N$, a spaced protocol will comprise a mixture of LTP1 and LTP2 (LTP$_\gamma$), since the initial train will induce LTP1 upon which subsequent trains will add LTP2 under our experimental conditions. The relative proportion of these two components will depend on a variety of conditions, including the interval between the trains, with ~10 min being optimal for the induction of LTP2[26].

**On the mechanism of LTP$_\gamma$.** The increase in $\gamma$ is regulated by the c-terminal tail of GluA1[48] and could result from a CaMKII-dependent phosphorylation of Ser831 of GluA1 to directly modulate their multiple conductance states[14,16] and/or by the insertion of CP-AMPARs[25], since these have a higher single channel conductance than CI-AMPARs[17,18]. Our findings have demonstrated that LTP$_\gamma$ can be explained exclusively by the latter mechanism, since all changes in $\gamma$ were eliminated by IEM. Furthermore, we found that activation of PKA plus CaMKII increased $\gamma$ whereas

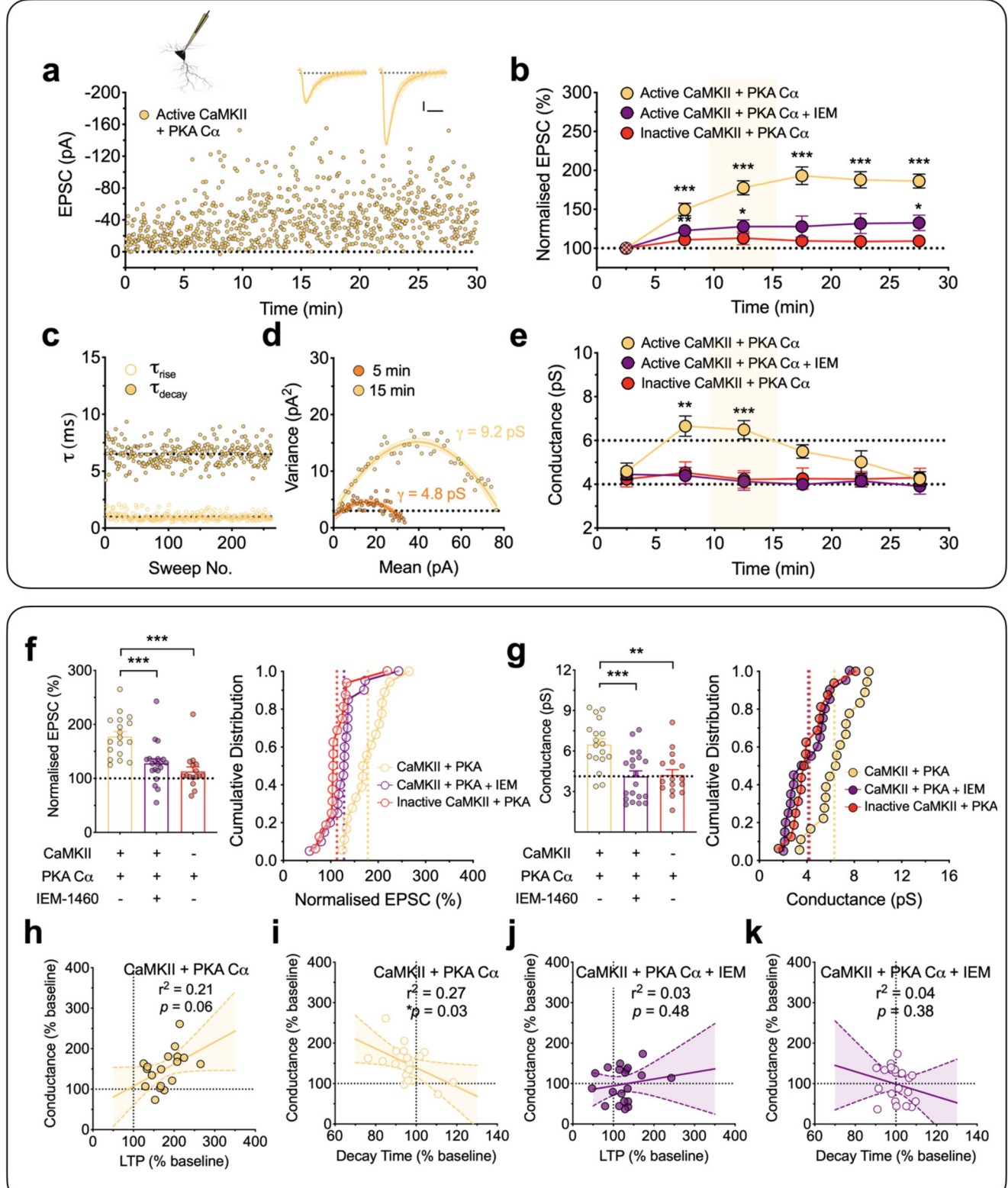

CaMKII alone did not, despite leading to a substantial potentiation. The failure of CaMKII alone to increase $\gamma$, which is contrary to some previous studies[14,15], could be explained on the basis of the native AMPAR configuration since $\gamma$ alterations are affected by the subunit combination and accessory protein composition of AMPA receptors[16,18]. It is worth noting, however, that whilst activation of CaMKII alone was not sufficient to induce LTP$_\gamma$ its

activation was necessary. It is possible, therefore, that phosphorylation of Ser831 of GluA1 is a necessary step for LTP$_\gamma$. Such a mechanism would involve dual phosphorylation of GluA1 on Ser831 and Ser845, which is known to occur[49]. One scenario is that GluA1 is firstly phosphorylated on Ser845 to drive CP-AMPARs to peri-synaptic sites, for which considerable evidence already exists[30,41,42,49–51]. From here, they are next

**Fig. 6 CaMKII plus PKA Cα results in a transient synaptic insertion of CP-AMPARs and increase in γ. a–k** Equivalent experiments as described in Fig. 5c–k but with the inclusion of activated CaMKII (250 U/mL) plus the catalytic subunit of PKA (PKA Cα, 300 U/mL) in the internal solution. Scale bars: 10 pA and 10 ms. **b** Pooled results (mean ± SEM, 5-min bins) for the effects on EPSC (%) by active CaMKII + PKA Cα ($n = 18$ neurons from 15 animals, $F_{(2.560,43.52)} = 38.34$, one-way repeated measures ANOVA), CaMKII + PKA Cα + IEM-1460 (IEM, 30 μM; $n = 20$ neurons from 15 animals, $F_{(2.064,39.22)} = 4.079$) and heat-inactivated CaMKII + PKA Cα ($n = 16$ neurons from 14 animals, $F_{(2.682,40.23)} = 1.301$). Bonferroni's post hoc multiple comparisons test; *$p < 0.05$, **$p < 0.01$ and ***$p < 0.001$ vs. the initial 5 min of whole-cell recording. **e** Time course for the estimates of γ. One-way repeated measures ANOVA followed by Bonferroni's multiple comparisons test (*$p < 0.05$, **$p < 0.01$ and ***$p < 0.001$ vs. the initial 5 min of whole-cell recording); $F_{(3.219, 54.72)} = 9.927$ (CaMKII + PKA Cα), $F_{(4.067,77.28)} = 0.5990$ (CaMKII + PKA Cα + IEM) and $F_{(3.587,53.80)} = 0.1366$ (heat-inactivated CaMKII + PKA Cα). Quantification for the levels of LTP ($F_{(2,51)} = 14.19$) (**f**) and γ ($F_{(2, 51)} = 9.210$) (**g**). One-way ANOVA followed by Bonferroni's multiple comparisons test; *$p < 0.05$, **$p < 0.01$ and ***$p < 0.001$. **h, i** Analysis of the relationships between γ and LTP ($p = 0.0618$, $F_{(1,16)} = 4.073$, F-test) (**h**) and EPSC decay time ($p = 0.0340$, $F_{(1,16)} = 5.440$, F-test) (**i**) for CaMKII + PKA Cα. **j, k** Analysis of the relationships between γ and LTP ($p = 0.4775$, $F_{(1,18)} = 0.5263$, F-test) (**j**) and EPSC decay time ($p = 0.3760$, $F_{(1, 18)} = 0.8241$, F-test) (**k**) for CaMKII + PKA Cα + IEM. Data are presented as mean ± SEM. Source data are provided as a Source Data file.

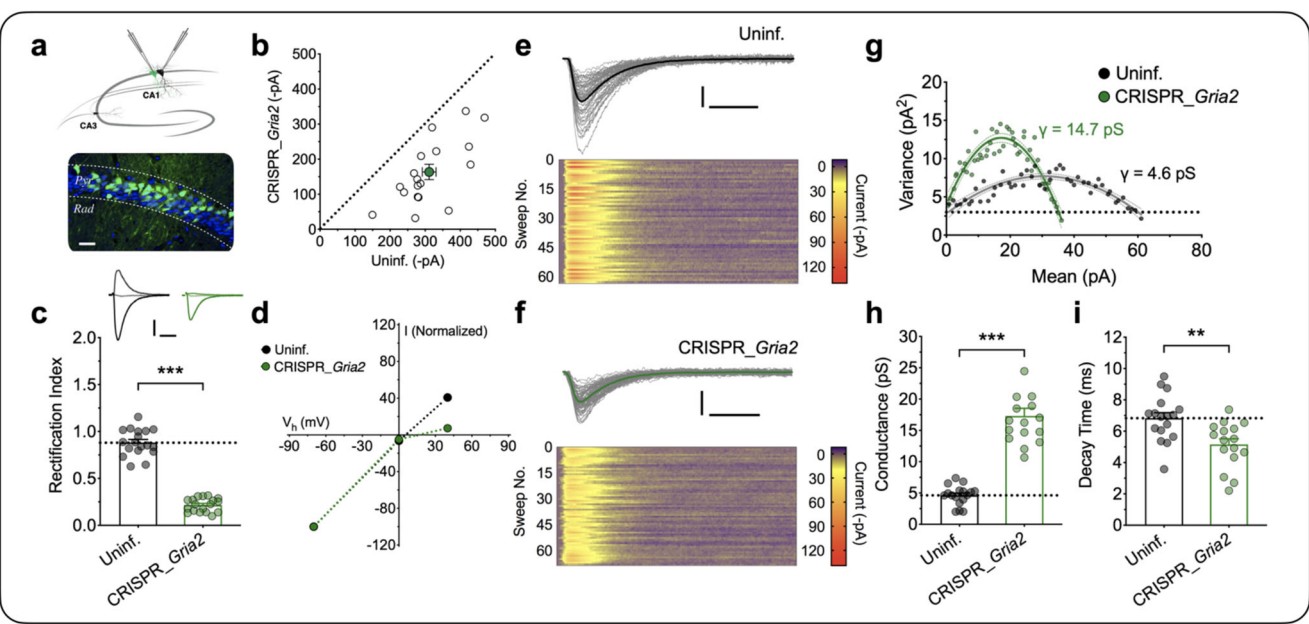

**Fig. 7 CP-AMPAR characterization in CRISPR_Gria2 knock-out neurons. a** Schematic of dual whole-cell recordings for the CRISPR_Gria2 knock-out and neighbouring uninfected (Uninf.) neurons. Sparse expression following stereotactic lentivirus injection detected by co-expressed EGFP (green); blue, DAPI staining; Pyr, stratum pyramidale; Rad, stratum radiatum. Scale bar = 30 microns. **b** Scatterplot shows amplitudes of AMPAR EPSCs for each pair recorded simultaneously (open circles) and the mean ± SEM (filled circle; $n = 18$ pairs from 12 animals). **c, d** Quantification of the rectification index for pharmacologically isolated AMPAR-mediated EPSCs and the corresponding current–voltage relationship (mean ± SEM, $n = 18$ pairs from 12 animals, $t_{17} = 17.38$, $p < 0.0001$, two-sided paired Student's t test). Scale bars: 100 pA and 10 ms. **e–g** Representative traces to measure the γ for the control and CRISPR_Gria2 knock-out neurons. Individual traces (thin lines) superimposed with the average. Scale bars: 30 pA and 10 ms. The lower panels are corresponding colour-coded images of all sweeps used in the NSFA (**g**). **h, i** Quantification of γ ($t_{31} = 10.09$, $p < 0.0001$, two-sided unpaired Student's t test) and decay time ($t_{31} = 3.273$, $p = 0.0026$, two-sided unpaired Student's t test) constants ($n = 17$ and 16 for Uninf. vs. CRISPR_Gria2 knock-out neurons from 12 animals). Data are presented as mean ± SEM. Source data are provided as a Source Data file.

phosphorylated on Ser831 to drive them into the synapse. In this model, both phosphorylation steps are required for synaptic γ to increase because they are regulating different trafficking steps on route to the synapse. In which case, CaMKII should not be able to increase γ further in neurons lacking GluA2 because CP-AMPARs are already synaptically expressed. Future work could address this and other aspects of the temporal sequence and consequences of PKA and CaMKII-dependent phosphorylation of GluA1 for LTP.

Our data are compatible with an exchange of a subset of CI-AMPARs for CP-AMPARs. The latter could be explained by a mechanism involving the $Ca^{2+}$ sensor PICK1[52,53], which has been shown to bind and internalize GluA2-containing AMPARs to enable the insertion of CP-AMPARs during LTP[54]. The next step involves the replacement of the newly inserted CP-AMPARs with CI-AMPARs, a process that requires baseline (low frequency) synaptic activation[26,43] and probably involves $Ca^{2+}$

permeation through the CP-AMPARs themselves[55]. The rapid replacement of CP-AMPARs with CI-AMPARs was originally described at excitatory synapses onto cerebellar stellate neurons from P18-P20 rats[56]. At this synapse, high-frequency stimulation (tetanus) induces CP-AMPARs to be replaced with the equivalent number of CI forms resulting in a reduction in the synaptic current by a third, reflecting lower γ of the latter form. We observed an initial reduction in EPSC amplitude following the triggering of LTP, which might be explained, in part, by a one-to-one exchange of CP-AMPARs for CI-AMPARs. Additionally, the transient expression of CP-AMPARs could trigger an increase in the number of AMPAR slots at synapses that enables an increase in the number of CI-AMPARs above and beyond what can occur during LTP1.

Since CP-AMPARs increase synaptic conductance why does there need to be an exchange for a greater number of CI-

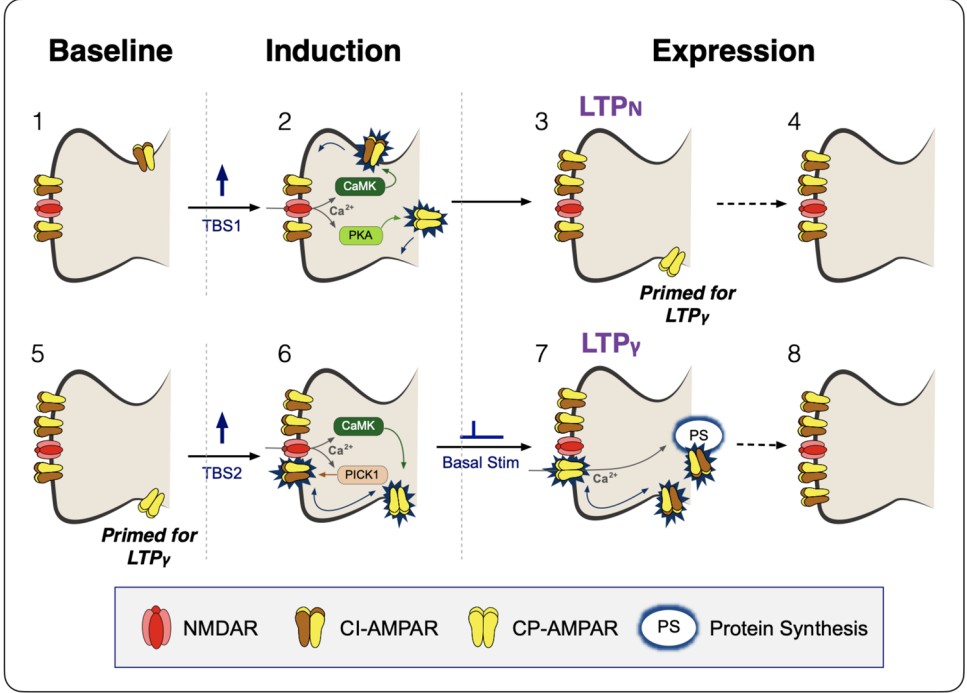

**Fig. 8 Schematic outlining the induction of two mechanistically distinct forms of LTP. 1** Under baseline conditions synaptic transmission is mediated by GluA2-containing, calcium-impermeable (CI)-AMPARs, two shown for simplicity. **2** The first theta-burst stimulation (TBS) activates NMDA receptors (NMDARs) and this drives more CI-AMPARs into the synapse by lateral diffusion from a peri-synaptic pool, via a process that involves CaMKII. PKA is also activated (via adenyl cyclase, not shown) and this induces the process of inserting GluA2-lacking calcium-permeable (CP)-AMPARs into peri-synaptic sites on the plasma membrane. **3** LTP is expressed by the increase in number of CI-AMPARs (LTP$_N$) but synapses also become primed for LTP$_\gamma$ by the availability of peri-synaptic CP-AMPARs. **4** Within ~1 h, the peri-synaptic CP-AMPARs are removed and, presumably, degraded. **6** If a second TBS is delivered whilst the synapses are still primed (**5**) then NMDAR activation drives the peri-synaptically located CP-AMPARs into the synapse, via a CaMKII-dependent process. This might involve an exchange of CP-AMPARs for CI-AMPARs, which are removed from the synapse via a mechanism triggered by PICK1. **7** These CP-AMPARs increase synaptic strength due to their higher single channel conductance (LTP$_\gamma$). However, their dwell time in the synapses is quite short (~15 min) before they are removed. If synapses remain active, such as by basal stimulation, activation of the transiently available, synaptic CP-AMPARs triggers protein synthesis and the insertion of more CI-AMPARs (**8**), which can extend the expression of LTP for long periods.

AMPARs to maintain the enhanced synaptic response? One possibility is that the expression of CP-AMPARs at these synapses needs to be restricted in time due to potential excitotoxicity[57]. Therefore, they can only provide a transient mechanism of expression whilst triggering the more persistent switch resulting in a larger number of CI-AMPARs.

**Developmental regulation of the expression mechanisms of LTP.** There is strong evidence that the expression mechanisms of LTP are developmentally regulated. The co-existence of two mechanisms involving the insertion of CI-AMPARs and CP-AMPARs can account for the LTP at P14[6] and in young adults, as observed herein. However, at around P7, LTP is associated with a decrease in $\gamma$[22], which is most likely explained by the replacement of CP-AMPARs with a larger number of CI-AMPARs. Early in development there is also the initial insertion of CP-AMPARs into synapses[55] that appear to lack AMPARs altogether; so-called "silent" synapses[58,59]. A potential scenario is as follows: first synapses acquire CP-AMPARs, next these are replaced by more CI-AMPARs. Thereafter LTP can increase the number of these CI-AMPARs via two mechanisms, one of which involves the transient insertion of CP-AMPARs and one that does not.

There have been far fewer studies regarding the mechanisms of synaptic plasticity in tissues obtained from adult animals compared to juvenile animals, although most learning and memory studies are conducted in adult animals. This is a concern when attempting to relate mechanisms of synaptic plasticity to learning and memory. Our present study, conducted exclusively

in tissue from young adult animals, shows that two distinct forms of synaptic plasticity can be readily induced simply by altering the patterns of activation. Our result that a cTBS protocol induces LTP that does not involve an alteration in $\gamma$ is consistent with another study in adult animals[24]. Our finding that a sTBS induces an additional component of LTP that involves an increase in $\gamma$ is the first evidence that such a process occurs beyond early developmental stages.

**Functional significance of two forms of LTP.** This raises the question as to why there are two distinct mechanisms to increase the synaptic complement of CI-AMPARs. Previous work has shown that the insertion of CP-AMPARs is specifically associated with the PKA and protein synthesis component of LTP[27]. It is reasonable to assume, therefore, that the transient insertion of CP-AMPARs is part of the machinery that triggers de novo protein synthesis and the consequential morphological changes (spine enlargement and/or new spine formation). In contrast, in the absence of de novo protein synthesis, the increase in synaptic CI-AMPAR number can support increased synaptic efficacy. Although both processes can increase synaptic strength lasting many hours in vitro, it seems probably that only the protein synthesis-dependent form triggers synaptic changes that underpin long-lasting memories (lasting from days to lifetimes). Indeed, it has been shown that spaced training with access to reward enhances the persistence of memory, and treatment with rolipram after training enhances memory retention[60]. It seems likely that PKA triggers protein synthesis by phosphorylating

GluA1 on S845 to promote the insertion of CP-AMPARs and by phosphorylating other regulatory targets and that together these regulate gene expression. The requirement for PKA to trigger the protein synthesis-dependent form of LTP also provides the opportunity for extensive neuromodulation. Neurotransmitters, such as noradrenaline and dopamine, and stress hormones, such as corticosterone, may, via the insertion of CP-AMPARs, augment protein synthesis-dependent LTP to enhance and/or prolong the persistence of the associated memory (e.g., [30,31,42,60–62]).

**Concluding remarks**. We have identified the molecular basis of two independent forms of LTP that co-exist at hippocampal synapses in young adult animals, the occurrence of which is controlled by the patterns of synaptic activation during induction. The existence of these two distinct LTP mechanisms goes a long way in explaining many of the controversies that have plagued the field. LTP1 can be induced by a cTBS and involves the insertion of CI-AMPARs, and for this to occur activation of CaMKII is both necessary and sufficient. A sTBS, however, triggers both LTP1 and LTP2. This latter form of LTP involves the transient insertion of CP-AMPARs and this requires activation of PKA in addition to CaMKII.

## Methods

**Hippocampal slice preparation**. Transverse hippocampal slices (400 μm) were prepared from male Sprague-Dawley rats (1–3 months of age). Animals were anesthetized with isoflurane and euthanised by decapitation in accordance with UK Animals (Scientific Procedures) Act of 1986. The brain was then removed and placed in ice-chilled slicing solution that contained (mM): 124 NaCl, 3 KCl, 26 NaHCO₃, 1.25 NaH₂PO₄, 10 MgSO₄, 10 D-glucose and 1 CaCl₂, saturated with 95% O₂ and 5% CO₂. The hippocampi were rapidly isolated from the brain and sliced using a vibratome (Microslicer) while maintained in the slicing solution. The CA3 region was removed to suppress the upstream neuronal excitability, and the slices were transferred to an incubation chamber that contained the recording solution (artificial cerebrospinal fluid, ACSF; mM): 124 NaCl, 3 KCl, 26 NaHCO₃, 1.25 NaH₂PO₄, 2 MgSO₄, 10 D-glucose and 2 CaCl₂ (carbonated with 95% O₂ and 5% CO₂). Slices were allowed to recover at 32–34 °C for 30 min, and then maintained at 26–28 °C for a minimum of 1 h before recordings were made.

**Field excitatory postsynaptic potential (fEPSP) recordings**. The extracellular electrophysiology was performed in both interface and submerged type chambers maintained at 32 °C, and continuously perfused at 2–4 mL/min with oxygenated ACSF. The slope of evoked fEPSPs (V/s) was measured in the CA1 region of hippocampal slices and bipolar stimulating electrodes were used at a constant voltage intensity (0.1 ms pulse width) throughout the experiments. Signals were amplified using Axopatch 1D (Molecular Devices) and digitized with BNC-2110 (National Instruments) A/D board at a sampling rate of 20 kHz. Recordings were monitored and analyzed using WinLTP v2.3[63]. Each specific experiment was conducted on a single slice from an animal, so the n-value reflects both the number of slices and animals used.

Two independent Schaffer collateral-commissural pathways (SCCPs) were stimulated alternately to obtain the evoked synaptic responses, each at a constant baseline frequency of between 0.033 and 0.1 Hz. Following a stable baseline period of at least 20 min, LTP was induced using theta-burst stimulation (TBS) delivered at the same basal stimulus intensity. An episode of TBS comprises 5 bursts at 5 Hz, with each burst composed of 5 pulses at 100 Hz. For LTP induced by compressed TBS (cTBS), three TBS episodes were delivered with an inter-episode interval (IEI) of 10 s. For spaced TBS (sTBS), the same number of episodes were given with an IEI of 10 min (see Fig. 1b). Representative sample traces are an average of 5 consecutive responses, collected from typical experiments (stimulus artefacts were blanked for clarity).

**Whole-cell patch clamp recording**. Whole-cell recording was made with ACSF that contained 50 μM picrotoxin (Abcam) and 20 μM (+)-bicuculline (Hello Bio) to prevent GABA_A receptor mediated contribution. CA1 pyramidal cells were visualized with IR-DIC optics (Zeiss). The whole-cell solution comprised (mM): 8 NaCl, 130 CsMeSO₃, 10 HEPES, 0.5 EGTA, 4 Mg-ATP, 0.3 Na₃-GTP, 5 QX-314 and 0.1 spermine. The pH was adjusted to 7.2–7.3 with CsOH and osmolarity was set to 285–290 mOsm/L. The peak amplitude of evoked EPSCs (pA) was monitored and analyzed using WinLTP v2.3[63]. Two independent SCCPs were stimulated alternately, each at a baseline frequency of 0.1–0.5 Hz. Borosilicate glass pipettes were fire-polished with a final resistance of 2–4 MΩ. Access resistance ($R_A$) was estimated by fitting whole-cell capacitance current with a double exponential, and experiments were only accepted for analysis if $R_A$ varied by <15%. $R_A$ values were

8.8 ± 0.3 MΩ; range from 6.2 to 12.8 MΩ. Signals were amplified using an Axopatch 200B (Molecular Devices), filtered at 2–5 kHz, and digitized at 20 kHz using a BNC-2110 (National Instruments) A/D board.

Cells were voltage-clamped at −70 mV throughout unless otherwise indicated. LTP was induced using TBS delivered at basal stimulus intensity while in current-clamp mode, and was triggered within 10 min of whole-cell to prevent the dialysis effect. In some experiments, the PKA catalytic subunit (PKA Cα, 300 U/mL) and/or CaMKII (250 U/mL) were included in the internal solution. CaMKII was activated (1× NEBuffer for Protein Kinases; 50 mM Tris-HCl, 10 mM MgCl₂, 0.1 mM EDTA, 2 mM DTT and 0.01% Brij 35; 200 μM ATP, 1.2 μM calmodulin and 2 mM CaCl₂; incubated for 10 min at 30 °C) or heat-inactivated (65 °C for 20 min) as described in the suppliers' manual (New England Biolabs). It is a Ca²⁺/calmodulin-dependent, truncated monomer (1–325 amino acid residues) of the α subunit, isolated from Spodoptera frugiperda (Sf9) cells infected with recombinant baculovirus carrying the truncated rat CaMKII (New England Biolabs).

To ensure recording stability, extracellular field EPSPs were simultaneously monitored as described previously[26]. Peak amplitude (pA) and initial slope (V/s) of EPSCs and fEPSPs were measured, and displayed on-line, using WinLTP v2.3[63]. Whole-cell recordings were initiated following collection of at least 10 min of stable baseline assessed by extracellular recordings.

**Peak-scaled, non-stationary fluctuation analysis (NSFA)**. The unitary conductance (γ) of AMPA receptors was estimated using NSFA according to ref. [6] (see also [19–21]). Whole-cell responses were carefully selected for analysis using WinWCP v5.1 (University of Strathclyde, Glasgow) and Mini Analysis v6.0 (Synaptosoft) software on the basis of the following criteria: first, precise alignment of traces on the rise phase; second, no contamination by spontaneous or polysynaptic currents; third, complete decay from the peak EPSCs. The traces were analyzed and the variance of the decay was plotted as a function of the amplitude at that time point. The x-axis was divided into 50-bins of equal current decrement from the peak. The single channel conductance was estimated by fitting the plot to a second polynomial equation, $σ² = iI - I²N + b_1$, where $σ²$ is the variance, $I$ is the mean current, $N$ is the number of channels activated, $i$ is the single channel current and $b_1$ is the background noise. In the conductance conversion (i.e. $γ = i/V$), the driving force ($V$) is the difference between the holding (−70 mV) and reversal potential (assumed to be 0 mV).

The kinetics of the mean EPSC from each neuron was estimated in Clampfit v10.1 (Molecular Devices) by measuring 20–80% rise time ($τ_{rise}$) and the time constant for the decay ($τ_{decay}$). Representative sample traces are the averages of all of the traces that were selected for analysis, superimposed with individual peak-scaled traces (10 successive sweeps), unless otherwise stated. Stimulus intensity was set to obtain a sporadic observation of transmission failures but high enough to obtain a reliable estimate of γ.

**Plasmid constructs and lentivirus production**. The following oligonucleotide sequences were used to generate single guide RNA (sgRNA) for GluA2 knockout: forward (5′ to 3′) CACC G ctaacagcatacagataggt; reverse (5′ to 3′) AAAC acctatctgtatgctgttag C[64]. These were annealed and ligated into the lentivirus backbone developed by the Zhang lab[65]. The construct was modified and used with the CaMKIIα promoter for Cas9-P2A-EGFP expression.

Lentivirus was produced by transfecting Lenti-X 293 T cells (Takara Bio) with pMD2.G, psPAX2 and lentiCRISPR[65]. The 293 T cells were maintained in serum-free UltraCULTURE media (supplemented with 4 mM L-glutamine, 2 mM GlutaMAX-I, 0.1 mM MEM non-essential amino acids, 1 mM sodium pyruvate, 1× penicillin/streptomycin). Three days after transfection, the supernatant was filter sterilized (0.45 μm pore membrane, Millipore) and ultracentrifuged at 110,000 × g (Beckman Coulter) with an additional sucrose filtration. The lentivirus pellet was resuspended in Dulbecco's PBS and kept at −80 °C.

**In vivo stereotactic injections and dual whole-cell recordings**. The surgical procedure was performed under sterile conditions in accordance with the Institutional Animal Care and Use Committee of Seoul National University. Male C57BL/6 mice (2–3 months of age) were anesthetized by intraperitoneal injection of a ketamine (130 mg/kg body weight) and xylazine (10 mg/kg) mixture. The anesthetized mice were immobilized on a stereotactic apparatus and the lentiviral medium (0.5 μL per each at a flow rate of 0.1 μL/min; 5 × 10⁹ TU/ml) was bilaterally injected at CA1 area using a microinjection syringe (Hamilton). The coordinates used were −1.7 mm posterior, ±1.2 mm lateral to bregma and −1.5 mm below the skull surface.

Following 4–6 weeks of expression, the hippocampal slices were prepared and whole-cell recordings were made as described above. EGFP-positive and neighbouring uninfected neurons were identified by epifluorescence microscopy and compared by dual whole-cell recordings. Rectification index was measured as described in ref. [26]. AMPAR currents were isolated using a mixture of D-AP5 (100 μM) and L-689,560 (5 μM). The index was calculated by taking the responses from −70, 0 and +40 mV of holding voltages. Following the recordings, brain slices were PFA-fixed, stained with DAPI, and imaged on a confocal microscope (Leica SP8).

**Compounds**. Drugs were prepared as frozen stock solutions (stored below −20 °C). Compounds were as follows: *N,N,H,*-Trimethyl-5-[(tricyclo[3.3.1.13,7]dec-1-ylmethyl)amino]-1-pentanaminium bromide hydrobromide (IEM-1460; Hello Bio); 4-(3-(cyclopentyloxy)-4-methoxyphenyl)pyrrolidin-2-one (rolipram; Abcam); 4-[(2 S)-2-[(5-isoquinolinylsulfonyl)methylamino]-3-oxo-3-(4-phenyl-1-piperazinyl)propyl] phenyl isoquinolinesulfonic acid ester (KN-62; Tocris and Hello Bio); D-AP5 (Hello Bio); L-689,560 (Tocris); a catalytic subunit of protein kinase A (PKA Cα, New England Biolabs); Ca$^{2+}$/calmodulin-dependent protein kinase II (CaMKII, New England Biolabs).

**Statistical analysis**. All treatment groups were interleaved with control experiments. Data are presented as mean ± SEM (standard error of the mean). Responses were normalized to the baseline prior to LTP induction unless otherwise stated. Statistical significance was assessed using (two-tailed) paired or unpaired Student's *t* tests or one-way ANOVA as appropriate using Graphpad Prism 8. Adjustments were made for multiple comparisons using Bonferroni's correction. The level of significance is denoted on the figures as follows: $*p < 0.05$, $**p < 0.01$ and $***p < 0.001$.

**Reporting summary**. Further information on research design is available in the Nature Research Reporting Summary linked to this article.

## Data availability

All original data are available upon reasonable request from the authors; the values for data underlying the Figures are provided as a Source Data file. Source data are provided with this paper.

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

## Acknowledgements

This work was supported by the MRC, ERC, CIHR Foundation Grant #154276 (G.L.C.) and Brain Canada Foundation (G.L.C.) through the Canada Brain Research Fund, with the financial support of Health Canada. G.L.C. is the holder of the Krembil Family Chair in Alzheimer's Research. This work was also supported by The National Honor Scientist Program (NRF-2012R1A3A1050385) of Korea (B.-K.K.).

## Author contributions

P.P. and G.L.C. conceived the study. P.P. carried out the experimental work. K.-H.K. and J.-i. K. helped with CRISPR experiments. T.M.S., Z.A.B., H.K., M.Z. and B.-K.K. provided advice and technical and/or financial support. P.P., J.G., C.A.B. and G.L.C. wrote the manuscript.

## Competing interests

The authors declare no competing interests.
