## [Peer Review File · Nature Communications]

REVIEWER COMMENTS

Reviewer #1 (Remarks to the Author):

In this study the authors investigate the contribution calcium permeable (CP) AMPA receptors play in LTP produced by two different LTP induction protocols.

Here a compressed Theta burst protocol (cTBS) is shown to produce robust LTP but no change AMPA receptor single channel conductance (γ). In contrast, a spaced TBS (sTBS) protocol produced LTP but was accompanied by a significant and transient increase in γ . Increases in γ are consistent with increased contribution from CP-AMPA receptors. This change in γ was observed using a weak TBS (wTBS) protocol when PKA activation was stimulated by rolipram or PKA Ca. Blocking CP-AMPA receptors with IEM-1460 prevented increased γ by wTBS with cellular PKA-Ca infusion and reduced LTP magnitude by 27%. The authors go on to show that cellular infusion of both active CaMKII and PKA-Ca together (but neither alone) is sufficient to produce increases in γ . Based on changes in γ in GluA2-lacking neurons it is estimated that $\sim 30\%$ of the synaptic current is from CP-AMPA receptors when changes in γ are observed following LTP induction.

Major issues:

- 1) The question of whether CP-AMPA receptors are inserted into hippocampal synapses during LTP is an interesting one and the study is generally well executed. However, the present study represents a rather incremental follow up to previous work from this group (Park et al. *Journal of Neuroscience*, 2016). In the 2016 paper this group shows that sTBS but not cTBS produces an increase in the rectification index of AMPA receptors. Rectification is another hallmark of CP-AMPA receptors, and this change is also tied to PKA in Park et al, 2016. Thus, the broader conclusions drawn in the present study using γ were previously drawn measuring AMPA receptor rectification. The present study does go further showing higher time resolution of the transient insertion of CP-AMPA receptors and that cellular infusion of active CaMKII and PKA-Ca together produce this change in γ but such advances are rather modest and are better suited for a more specialized journal (e.g. *J Neurosci*).
- 2) Given that IEM-1460 has been shown to inhibit NMDA receptors it is important to directly test whether $30\mu\text{M}$ IEM-1460 has any impact on NMDA receptor function in the authors' slice preparation. A small reduction in NMDA receptor function might account for the reduction in LTP magnitude shown in Fig 4g.
- 3) The authors argument that sLTP dependent changes in γ are not mediated by GluA1 phosphorylation would be significantly strengthened by showing that sLTP dependent changes in γ are not observed in neurons lacking GluA2.

Reviewer #2 (Remarks to the Author):

In the current work, Park et al., examine the detailed mechanisms of AMPA receptor mediated synaptic potentiation in Schafer collateral-CA1 long-term potentiation (LTP) using slice physiology. The key question addressed is whether AMPAR mediated LTP occurs via an increase in the number of AMPARs or a change in the single channel conductance. The work uses several different strategies but relies in particular on non-stationary fluctuation analysis (NSFA). The simultaneous use of field recording and single-cell patch clamp recording is a major strength. The authors show the LTP can be mediated by increased AMPAR numbers or conductance, depending on the precise induction method used. The authors also show that these distinct AMPAR mediated LTP mechanisms have different reliance on CaMKII and PKA. Increased AMPAR conductance requires both PKA and CaMKII and is mediated by calcium-permeable AMPARs. The type of LTP induction that recruits CP-AMPARs requires

protein synthesis based on prior work. The authors provide exciting speculation that the recruitment of CP-AMPA receptors may initiate the protein synthesis needed to stabilize the plasticity and memory. This exciting speculation provides an important direction for future work.

This area of research is very mature and a great deal of the molecular mechanisms of AMPAR-mediated LTP in hippocampal CA1 neurons have been elucidated. However, there are several long-standing controversies, particularly regarding the role of CP-AMPA receptors. Multiple studies going back over more than a decade go back and forth on this matter. Key issues that have been raised are the developmental age and the precise induction methods used. The work presented here is of exceptional quality. These experimental methods are very demanding. The authors provide very detailed quantitative comparisons of LTP induced by distinct paradigms and elucidate specific signaling pathways. This work convincingly shows that protein synthesis-dependent forms of hippocampal LTP do indeed recruit CP-AMPA receptors in adults via activation of PKA. This work is a very important contribution in that it will go a long way to settle a long-standing debate in this field, and therefore is likely to be of high interest to many readers. I do not have any major concerns over the quality of the data or the strength of the conclusions.

Comment

The authors clearly show that both CaMKII and PKA are required for the recruitment of CP-AMPA receptors and for the transient increase in channel conductance during sTBS. In the discussion the authors specifically mention that GluA1 is phosphorylated by CaMKII on S831 but they fail to mention direct phosphorylation of S845 by PKA. In my opinion it would be appropriate to add this point to the discussion as it is certainly relevant to the conclusions of this work. Moreover, S845 phosphorylation by PKA has been shown to be crucial for the synaptic recruitment of CP-AMPA receptors in multiple studies. With the current data presented the authors have a chance to also engage another debate in the field. In a recent paper, Hosokawa et al., *Neuron* 2015 argued that AMPAR phosphorylation was very limited and that GluA1 was never "dual-phosphorylated" on S831 and S845, despite a large body of prior research supporting this idea. This current work clearly shows coordinated activities of CaMKII and PKA in mediating LTP via increased AMPAR conductance. This issue could be raised in the discussion. A recent study demonstrated that "dual phosphorylation" does indeed occur, Diering et al., *PNAS* 2016. Coordinated post-translational modifications of AMPARs was recently reviewed Diering and Huganir, *Neuron* 2018. Coordinated phosphorylation of GluA1 is certainly relevant to the current findings. While this is not a crucial point, the discussion would be enriched by some addition addressing these points.

Graham H. Diering

Reviewer #3 (Remarks to the Author):

The authors provide convincing evidence based on several lines of investigations that activation of PKA promotes temporary incorporation of GluA2-lacking CP-AMPA receptors at postsynaptic sites during certain forms of LTP. Importantly, this effect is strictly correlated with a temporary increase in single channel conductance (γ), which is at least in part because CP-AMPA receptors have a higher conductance than the predominant GluA2-containing CI-AMPA receptors. The findings of temporary insertion of CP-AMPA receptors are consistent with earlier work and the findings of temporary increase in γ provides an important expansion of this earlier work.

In more detail, induction of LTP with three episodes of spaced (10 min apart) but not compressed (10 s apart) TBS resulted in PKA-dependent temporary insertion of CP-AMPA receptors as indicated by both, temporary increase in rectification and in γ . The sensitivity of all of these effects to IEM1460 (which inhibits CP-AMPA receptors but not CI-AMPA receptors) indicates that the increase in LTP is a result of the temporary insertion of CP-AMPA receptors. It appears likely that compressed LTP does not provide sufficient time for CP-AMPA receptors to make it in time to the postsynaptic site to augment postsynaptic Ca influx during the second and third episodes of TBS. Injection of CaMKII plus PKA versus CaMKII alone mimicked spaced versus compressed LTP, which is a remarkable finding in terms of contribution by PKA to augment LTP.

This MS is well developed with multiple controls and consistent findings between different experimental approaches. Thus, I only have modest and minor concerns.

Modest/Minor Concerns

1. Fig. 5A,B. KN62 has several side effects and is not a 'clean' CaMKII inhibitor-ideally another more specific inhibitor such as myristoylated or tat-tagged AIP or CN21 peptides would be used to confirm the role of CaMKII in cLTP and sLTP. However, this is only a modest concern because injection of CaMKII mimics LTP in whole cell recordings.
2. Fig. 5 C-K. More details on activation of CaMKII are required-was the whole activation mix injected including the Ca and calmodulin, which could cause some issues with respect to specificity of CaMKII action versus action of Ca or calmodulin? What was the source of CaMKII (expression of recombinant CaMKII or purified from native tissue)? Reference to Supplier's manual is not sufficient here because CaMKII autophosphorylated of T286 versus T305/T306 has different effects on AMPAR activity according to work by the late John Lisman (two papers around 2010). Ideally, the phosphorylation status of T286 and T305/T306 would be analyzed by immunoblotting with commercially available phosphospecific antibodies.
3. Fig. 4. TBS paired with PKA injection leads to strong LTP. Ideally the effect of just injecting PKA would be presented (without TBS). However, again this is a minor issue because Fig. 6 shows that combining PKA with inactive CaMKII does not affect EPSC (no potentiation or changes in gamma). Arrows to mark when TBS was applied are lacking in panel A and the time schedule here is not clear enough with respect to LTP induction versus when effects were recorded.
4. Discussion. The authors claim that their results can explain the disparate findings by Plant et al (Ref 43), who found temporary postsynaptic recruitment of CP-AMPA during LTP, and Adesnik et al. (Ref 44), who was not able to detect such recruitment, because, so this new work, different LTP induction protocols result in respective different outcomes. However, the findings by the authors cannot explain the difference because Plant et al and Adesnik et al used comparable induction protocols. What is more likely is that the exact ages (both groups state that they used 2-3 week old animals) vary such that Plant et al might have used closer to 2 weeks and Adesnik et al closer to 3 weeks. LTP induced by a single tetanus has been shown by Lu et al (2007; EMBO J 26, 4879-4890) to require PKA and CP-AMPA at 2 weeks and then again later from 8 weeks on but not 3 weeks of age. The work of Lu et al. (2007) also explains why Gray et al (Ref 45) did not find requirement of CP-AMPA because inly LTP induced by a single train of 100 Hz but not by two trains of 100 Hz required CP-AMPA. The age dependency has been recently confirmed by Sanderson et al 2015 (which is cited as Ref 32 but in different context in the MS). In this context a recent paper by Grant and co-workers is worth mentioning because it shows that synapse formation is highest at 3 weeks of age when perhaps stabilization and potentiation are tuned such that weak induction protocols can support stabilization and potentiation of synapses such that PKA and CP-AMPA might not be necessary at this exact age (Cizeron et al., 2020: Science 269, 270-275).
5. The 'minimal' stimulation protocol the authors used might have recruited only stronger synapses. The authors might want to discuss (or actually determine experimentally) whether only stronger synapses show the LTP mechanism(s) they describe.
6. The authors state that space limitations prevent them from a more in depth discussion of the LTP mechanisms. Perhaps they could add a section "Supplemental Discussion" to Supplemental Material?

RESPONSES TO REVIEWER COMMENTS

Reviewer #1 (Remarks to the Author):

In this study the authors investigate the contribution calcium permeable (CP) AMPA receptors
play in LTP produced by two different LTP induction protocols.

Here a compressed Theta burst protocol (cTBS) is shown to produce robust LTP but no change
AMPA receptor single channel conductance (γ). In contrast, a spaced TBS (sTBS) protocol
produced LTP but was accompanied by a significant and transient increase in γ . Increases in γ
are consistent with increased contribution from CP-AMPA receptors. This change in γ was
observed using a weak TBS (wTBS) protocol when PKA activation was stimulated by rolipram
or PKA $C\alpha$. Blocking CP-AMPA receptors with IEM-1460 prevented increased γ by wTBS with
cellular PKA- $C\alpha$ infusion and reduced LTP magnitude by 27%. The authors go on to show that
cellular infusion of both active CaMKII and PKA- $C\alpha$ together (but neither alone) is sufficient to
produce increases in γ . Based on changes in γ in GluA2-lacking neurons it is estimated that
~30% of the synaptic current is from CP-AMPA receptors when changes in γ are observed
following LTP induction.

Major issues:

1) The question of whether CP-AMPA receptors are inserted into hippocampal synapses during
LTP is an interesting one and the study is generally well executed. However, the present study
represents a rather incremental follow up to previous work from this group (Park et al. Journal of
Neuroscience, 2016). In the 2016 paper this group shows that sTBS but not cTBS produces an
increase in the rectification index of AMPA receptors. Rectification is another hallmark of CP-
AMPA receptors, and this change is also tied to PKA in Park et al, 2016. Thus, the broader
conclusions drawn in the present study using γ were previously drawn measuring AMPA
receptor rectification. The present study does go further showing higher time resolution of the
transient insertion of CP-AMPA receptors and that cellular infusion of active CaMKII and PKA-
$C\alpha$ together produce this change in γ but such advances are rather modest and are better suited
for a more specialized journal (e.g. J Neurosci).

**We appreciate that the reviewer acknowledges the quality of our study but with all due**
**respect we would disagree on the significance of the advance. Synaptic plasticity at this**
**synapse has been extensively studied because of its relevance to learning and memory and**
**brain disorders, yet very important controversies remain. We consider it to be of extreme**
**importance to this crucial field to strive to resolve these issues, and that to substantiate**
**conclusions as rigorously as possible and to disseminate this finding as widely as possible is**
**what is required. Our work makes many mechanistic advances beyond verifying with**
**improved time resolution the role of CP-AMPA receptors in LTP. In particular, we**
**unambiguously demonstrate the necessity and sufficiency of PKA and CaMKII in both**
**forms of LTP. Of course, different readers will place different values on studies that**
**rigorously tackle old questions from new angles versus those that appear more novel (with**
**all the uncertainties that this often entails). We are heartened that the other two reviewers**
**acknowledge the importance and significance of our work and strongly recommend**
**publication in Nat Comm.**

2) Given that IEM-1460 has been shown to inhibit NMDA receptors it is important to directly
test whether 30 μ M IEM-1460 has any impact on NMDA receptor function in the authors' slice
preparation. A small reduction in NMDA receptor function might account for the reduction in
LTP magnitude shown in Fig 4g.

**We did not consider it likely that IEM-1460 is inhibiting LTP induced by a sTBS via**
**inhibition of NMDA receptors for three main reasons. Firstly, we reported previously that**
**in interleaved experiments IEM had no effect whatsoever on LTP induced by a cTBS - a**
**protocol that activated NMDARs essentially to the same extent as the sTBS (Park et al.**
**2016; 2018 Ref 26 & 27). Secondly, the synaptic potentiation observed during and**
**immediately after each of the TBS in a sTBS protocol reveals no effect of IEM (Park et al.**
**2016; Ref 26 & 27). Third, in the present study we could readily induce an IEM-sensitive**
**potentiation by effectively bypassing the activation of NMDA receptors using patch loading**
**of CaMKII and PKA. However, we acknowledge that the direct demonstration of the**
**effects of IEM-1460 on NMDA receptor-mediated synaptic transmission is a valuable**
**addition to the study. We therefore performed this experiment and found no effect of**
**IEM-1460 on NMDA receptor-mediated EPSCs, evoked during stimulation of Schaffer**
**collaterals either with single pulses or TBS. These new data are illustrated in a new Figure,**
**which we propose to include as supplementary information (Sup.Fig.1).**

3) The authors argument that sLTP dependent changes in γ are not mediated by GluA1
phosphorylation would be significantly strengthened by showing that sLTP dependent
changes in γ are not observed in neurons lacking GluA2.

**We appreciate this suggestion. Unfortunately, due to staff turnover and the disruption to**
**the research laboratory caused by Covid-19 we are not in a position where we can carry**
**out this experiment at the present time. We consider it unlikely that there would be an**
**increase in γ in GluA2 KO neurons, where the baseline γ is already very high, however we**
**will attempt these experiments when we are in a position to do so. In the meantime, we flag**
**up this possibility in a revised Discussion and hope that the Reviewer will be satisfied by**
**this approach.**

83 //

Reviewer #2 (Remarks to the Author):

In the current work, Park et al., examine the detailed mechanisms of AMPA receptor mediated
synaptic potentiation in Schaffer collateral-CA1 long-term potentiation (LTP) using slice
physiology. The key question addressed is whether AMPAR mediated LTP occurs via an
increase in the number of AMPARs or a change in the single channel conductance. The work
uses several different strategies but relies in particular on non-stationary fluctuation analysis
(NSFA). The simultaneous use of field recording and single-cell patch clamp recording is a major
strength. The authors show the LTP can be mediated by increased AMPAR numbers or
conductance, depending on the precise induction method used. The authors also show that these
distinct AMPAR mediated LTP mechanisms have different reliance on CaMKII and PKA.
Increased AMPAR conductance requires both PKA and CaMKII and is mediated by calcium-

permeable AMPARs. The type of LTP induction that recruits CP-AMPARs requires protein
synthesis based on prior work. The authors provide exciting speculation that the recruitment of CP-
AMPARs may initiate the protein synthesis needed to stabilize the plasticity and memory. This
exciting speculation provides an important direction for future work.

This area of research is very mature and a great deal of the molecular mechanisms of AMPAR
mediated LTP in hippocampal CA1 neurons have been elucidated. However, there are several
long-standing controversies, particularly regarding the role of CP-AMPARs. Multiple studies
going back over more than a decade go back and forth on this matter. Key issues that have been
raised are the developmental age and the precise induction methods used. The work presented
here is of exceptional quality. These experimental methods are very demanding. The authors
provide very detailed quantitative comparisons of LTP induced by distinct paradigms and
elucidate specific signaling pathways. This work convincingly shows that protein synthesis
dependent forms of Hippocampal LTP do indeed recruit CP-AMPARs in adults via activation of
PKA. This work is a very important contribution in that it will go a long way to settle a long-
standing debate in this field, and therefore is likely to be of high interest to many readers. I do
not have any major concerns over the quality of the data or the strength of the conclusions.

**We are delighted that this reviewer states that “This work is a very important contribution
in that it will go a long way to settle a long-standing debate in this field, and therefore is
likely to be of high interest to many readers”.**

*Comment.* The authors clearly show that both CaMKII and PKA are required for the recruitment
of CP-AMPARs and for the transient increase in channel conductance during sTBS. In the
discussion the authors specifically mention that GluA1 is phosphorylated by CaMKII on S831
but they fail to mention direct phosphorylation of S845 by PKA. In my opinion it would be
appropriate to add this point to the discussion as it is certainly relevant to the conclusions of this
work. Moreover, S845 phosphorylation by PKA has been shown to be crucial for the synaptic
recruitment of CP-AMPARs in multiple studies.

**We apologize for this omission. We did have a detailed paragraph about the S845
phosphorylation by PKA in an earlier draft but had to edit down our manuscript to comply
with the length requirements of the journal. Since we had already gone into this topic in
detail in our previous publication in J. Neurosci. we decided to focus more on other aspects
of the molecular mechanism. For a more rounded discussion we have now included a
mention of this together with citations.**

With the current data presented the authors have a chance to also engage another debate in the
field. In a recent paper, Hosokawa et al., Neuron 2015 argued that AMPAR phosphorylation was
very limited and that GluA1 was never “dual-phosphorylated” on S831 and S845, despite a large
body of prior research supporting this idea. This current work clearly shows coordinated
activities of CaMKII and PKA in mediating LTP via increased AMPAR conductance. This issue
could be raised in the discussion. A recent study demonstrated that “dual phosphorylation” does
indeed occur, Diering et al., PNAS 2016. Coordinated post-translational modifications of
AMPARs was recently reviewed Diering and Huganir, Neuron 2018. Coordinated
phosphorylation of GluA1 is certainly relevant to the current findings. While this is not a crucial

point, the discussion would be enriched by some addition addressing these points.

**We agree that this issue of dual phosphorylation of GluA1 is an important one.**

**Accordingly, we have included a mention of this point as suggested in a revised paragraph**
**in the Discussion.**

153 //

Reviewer #3 (Remarks to the Author):

The authors provide convincing evidence based on several lines of investigations that activation
of PKA promotes temporary incorporation of GluA2-lacking CP-AMPA receptors at postsynaptic sites
during certain forms of LTP. Importantly, this effect is strictly correlated with a temporary
increase in single channel conductance (γ), which is at least in part because CP-AMPA receptors
have a higher conductance than the predominant GluA2-containing CI-AMPA receptors. The findings
of temporary insertion of CP-AMPA receptors are consistent with earlier work and the findings of
temporary increase in γ provides an important expansion of this earlier work.

In more detail, induction of LTP with three episodes of spaced (10 min apart) but not
compressed (10 s apart) TBS resulted in PKA-dependent temporary insertion of CP-AMPA receptors as
indicated by both, temporary increase in rectification and in γ . The sensitivity of all of
these effects to IEM1460 (which inhibits CP-AMPA receptors but not CI-AMPA receptors) indicates that the
increase in LTP is a result of the temporary insertion of CP-AMPA receptors. It appears likely that
compressed LTP does not provide sufficient time for CP-AMPA receptors to make it in time to the
postsynaptic site to augment postsynaptic Ca influx during the second and third episodes of TBS.
Injection of CaMKII plus PKA versus CaMKII alone mimicked spaced versus compressed LTP,
which is a remarkable finding in terms of contribution by PKA to augment LTP.

This MS is well developed with multiple controls and consistent findings between different
experimental approaches. Thus, I only have modest and minor concerns.

**We are grateful that this reviewer acknowledges that our manuscript is “well developed**
**with multiple controls and consistent findings between different experimental approaches”.**

**Modest/Minor Concerns**

1. Fig. 5A,B. KN62 has several side effects and is not a ‘clean’ CaMKII inhibitor-ideally another
more specific inhibitor such as myristoylated or tat-tagged AIP or CN21 peptides would be used
to confirm the role of CaMKII in cLTP and sLTP. However, this is only a modest concern
because injection of CaMKII mimics LTP in whole cell recordings.

**We acknowledge that KN62 has several side effects but, as the reviewer points, our**
**demonstration of the role of CaMKII in LTP is confirmed by the addition of CaMKII.**

**Indeed, the role of CaMKII in LTP in our study is based almost entirely on the use of the**
**kinase per se (with inactivated kinase as the control). The inclusion of the CaMKII data set**
**was included mainly as an introduction to the experiments. Since it was repeating previous**
**work it could be omitted without impacting the study.**

2. Fig. 5 C-K. More details on activation of CaMKII are required-was the whole activation mix
injected including the Ca and calmodulin, which could cause some issues with respect to
specificity of CaMKII action versus action of Ca or calmodulin? What was the source of
CaMKII (expression of recombinant CaMKII or purified from native tissue)? Reference to
Supplier's manual is not sufficient here because CaMKII autophosphorylated of T286 versus
T305/T306 has different effects on AMPAR activity according to work by the late John Lisman
(two papers around 2010). Ideally, the phosphorylation status of T286 and T305/T306 would be
analyzed by immunoblotting with commercially available phosphospecific antibodies.

**We have added additional information as requested:**

**“It is a Ca²⁺/calmodulin-dependent, truncated monomer (1–325 amino acid residues) of**
**the α subunit, isolated from *Spodoptera frugiperda* (Sf9) cells infected with recombinant**
**baculovirus carrying the truncated rat CaMKII (New England Biolabs; kindly provided by**
**Dr. H. Shulman).”**

**The control was heat-inactivated CaMKII, which included the calcium and calmodulin.**

3. Fig. 4. TBS paired with PKA injection leads to strong LTP. Ideally the effect of just injecting
PKA would be presented (without TBS). However, again this is a minor issue because Fig. 6
shows that combining PKA with inactive CaMKII does not affect EPSC (no potentiation or
changes in gamma). Arrows to mark when TBS was applied are lacking in panel A and the time
schedule here is not clear enough with respect to LTP induction versus when effects were
recorded.

**The effect of injecting PKA without TBS is actually presented in Fig 4a,b (labelled control**
**input). We appreciate that this wasn't made clear and so we have modified the text**
**accordingly and have added the arrow markers to indicate where the TBS was applied.**

4. Discussion. The authors claim that their results can explain the disparate findings by Plant et al
(Ref 43), who found temporary postsynaptic recruitment of CP-AMPA during LTP, and
Adesnik et al. (Ref 44), who was not able to detect such recruitment, because, so this new work,
different LTP induction protocols result in respective different outcomes. However, the findings
by the authors cannot explain the difference because Plant et al and Adesnik et al used
comparable induction protocols. What is more likely is that the exact ages (both groups state that
they used 2-3 week old animals) vary such that Plant et al might have used closer to 2 weeks and
Adesnik et al closer to 3 weeks. LTP induced by a single tetanus has been shown by Lu et al
(2007; EMBO J 26, 4879-4890) to require PKA and CP-AMPA at 2 weeks and then again
later from 8 weeks on but not 3 weeks of age. The work of Lu et al. (2007) also explains why
Gray et al (Ref 45) did not find requirement of CP-AMPA because inly LTP induced by a
single train of 100 Hz but not by two trains of 100 Hz required CP-AMPA. The age
dependency has been recently confirmed by Sanderson et al 2015 (which is cited as Ref 32 but in
different context in the MS). In this context a recent paper by Grant and co-workers is worth
mentioning because it shows that synapse formation is highest at 3 weeks of age when perhaps
stabilization and potentiation are tuned such that weak induction protocols can support
stabilization and potentiation of synapses such that PKA and CP-AMPA might not be
necessary at this exact age (Cizeron et al., 2020: Science 269, 270-275).

**We thank the reviewer for these valuable insights into what may explain the earlier**
**controversies of the Plant and Adesnik studies. We have expanded our discussion on this**
**point accordingly.**

5. The ‘minimal’ stimulation protocol the authors used might have recruited only stronger
synapses. The authors might want to discuss (or actually determine experimentally) whether only
stronger synapses show the LTP mechanism(s) they describe.

**Given the much higher proportion of low P(r) synapses we think it is likely that our min**
**stim involved a mixture of high and low P(r) synapses, as it known to be the case for**
**fEPSPs. In this work we measured EPSCs and fEPSPs simultaneously and saw the same**
**effects irrespective of recording method. Accordingly, we think this is unlikely but agree**
**that it’s worth a mention, which we have now included.**

6. The authors state that space limitations prevent them from a more in depth discussion of the
LTP mechanisms. Perhaps they could add a section “Supplemental Discussion” to Supplemental
Material?

**Thank you for the suggestion. We’ve been able to incorporate the additional discussion**
**within the main text.**

REVIEWERS' COMMENTS

Reviewer #1 (Remarks to the Author):

I feel confident in saying that there are few researchers more sympathetic to older questions surrounding LTP than myself. I maintain my original concern regarding the potential impact of this study but acknowledge that such concerns are somewhat subjective. Given the enthusiasm of the other two reviewers I am happy to defer to the editors of Nature Communications on this point. The lack of impact of IEM-1460 on NMDA receptors is appreciated and is indeed a valuable addition to the study. It is unfortunate that γ in GluA2 lacking neurons cannot be investigated at this time, but I am sympathetic to the difficulties that COVID has created. I am now satisfied with the technical aspects of the study and support whatever decision the editors ultimately make regarding publication.

Reviewer #2 (Remarks to the Author):

The current work from Park et al is expertly done. I had only modest suggestions from their first submission and these have been appropriately addressed in the revised manuscript.

Reviewer #3 (Remarks to the Author):

This MS provides important new evidence in support of a model of LTP that under certain but not necessarily all conditions the temporary postsynaptic insertion of CP-AMPARs is required. Thus, it is a significant step forward towards understanding molecular mechanisms of synaptic plasticity and thereby synaptic functionality. The authors have addressed all concerns.

There are a couple of editorial issues at this point.

1. Acknowledgement of the source of CaMKII: the correct spelling of Dr. Schulman's name is with the letter "c" not "Shulman." Also Dr. Schulman left academic well over 10 years ago. It is unclear when and how he was the source – perhaps his affiliation should be stated – it was not New England Biolabs as far as I know.

2. The statement in the Discussion "when a single train (tetanus or TBS) is employed, the resultant LTP is independent of both PKA activation and de novo protein synthesis" is not completely accurate. There are examples like Ref. 46 (Lu et al., 2007) that find that a single 1 sec, 100 Hz tetanus does require PKA and CP-AMPAR. At the same time, I would not equate this LTP (which was induced by a 'weak' stimulus and might not last permanently) to LTP that depends on protein synthesis and can last for a very long time. Rather, LTP induction protocols that are weak like the wLTP in the current MS or as in Ref. 46 require PKA and CP-AMPAR perhaps because CaMKII is not fast enough to drive AMPARs to the surface (although CaMKII is, as is the case for most forms of LTP, still required for postsynaptic AMPAR accumulation). Here PKA and its phosphorylation site on GluA1 (S845) might be required to drive surface insertion of AMPARs in the early phase(s) of such forms of LTP. Long-lasting forms of LTP that require protein synthesis for them to last need PKA likely to drive gene expression but the PKA targets could be different from S845. At the same time it is possible that the regulation of gene expression could also involve CP-AMPAR and for that purpose S845 phosphorylation might be required although the CP-AMPARs might not have to show up at the postsynaptic site at all during those stronger protocols.

Johannes W. Hell

**RESPONSES TO REVIEWER COMMENTS, NCOMMS-20-26580A**

November 27th, 2020.

**Reviewer #3 (Remarks to the Author):**

There are a couple of editorial issues at this point.

1. Acknowledgement of the source of CaMKII: the correct spelling of Dr. Schulman's name is with the
letter "c" not "Shulman." Also Dr. Schulman left academic well over 10 years ago. It is unclear when and
how he was the source – perhaps his affiliation should be stated – it was not New England Biolabs as far
as I know.

**1. The CaMKII was obtained from New England Biolabs. We have removed mention of**
**Dr. Schulman, who was acknowledged in the NEB product datasheet.**

2. The statement in the Discussion “when a single train (tetanus or TBS) is employed, the resultant LTP is
independent of both PKA activation and de novo protein synthesis“ is not completely accurate. There are
examples like Ref. 46 (Lu et al., 2007) that find that a single 1 sec, 100 Hz tetanus does require PKA and
CP-AMPA. At the same time, I would not equate this LTP (which was induced by a ‘weak’ stimulus
and might not last permanently) to LTP that depends on protein synthesis and can last for a very long
time. Rather, LTP induction protocols that are weak like the wLTP in the current MS or as in Ref. 46
require PKA and CP-AMPA perhaps because CaMKII is not fast enough to drive AMPARs to the
surface (although CaMKII is, as is the case for most forms of LTP, still required for postsynaptic
AMPAR accumulation). Here PKA and its phosphorylation site on GluA1 (S845) might be required to
drive surface insertion of AMPARs in the early phase(s) of such forms of LTP.

Long-lasting forms of LTP that require protein synthesis for them to last need PKA likely to drive gene
expression but the PKA targets could be different from S845. At the same time it is possible that the
regulation of gene expression could also involve CP-AMPA and for that purpose S845 phosphorylation
might be required although the CP-AMPA might not have to show up at the postsynaptic site at all
during those stronger protocols.

**2. For the Reviewer's editorial concern regarding the role of PKA in single train LTP**
**(with respect to Ref.46 and any other instances), we appreciate that there are such cases**
**and that these may not necessarily invoke protein synthesis dependence. Our goal here was**
**only to introduce the definition of LTP1 (and then contrast to LTP2). Therefore, we**
**propose to insert the qualifier “may be” on line 285.**

**Furthermore, we have made an additional change later on, where we explain that there are**
**a number of factors that regulate CP-AMPA-dependent synaptic plasticity, such as**
**developmental stage and stress; we have now also expanded to say that the precise**
**stimulus parameters and other experimental conditions used are also factors (see**
**immediately preceding Ref.46). The sentence starting on line 313 now reads as follows:**

“However, the extent to which CP-AMPA are involved in synaptic plasticity is likely
to involve additional factors, such as the developmental stage of the animal, the level of
stress experienced prior to euthanasia and the precise experimental conditions used,
including the stimulus parameters employed^{31,32,43,44,46,47}.”

**Regarding the point about the potential underlying mechanism we have added the**
**following sentence on line 410:**

“It seems likely that PKA triggers protein synthesis by phosphorylating GluA1 on S845
to promote the insertion of CP-AMPA receptors and by phosphorylating other regulatory targets
and that together these regulate gene expression.”

**We appreciate all the reviewer comments that have helped improve our manuscript.**